# High resolution vertical distribution and sources of HONO and NO$_2$ in the nocturnal boundary layer in urban Beijing, China

**Fanhao Meng**[1,2], **Min Qin**[1], **Ke Tang**[1,2], **Jun Duan**[1], **Wu Fang**[1], **Shuaixi Liang**[1,2], **Kaidi Ye**[1,2], **Pinhua Xie**[1,2,3,5], **Yele Sun**[3,4,5], **Conghui Xie**[4], **Chunxiang Ye**[6], **Pingqing Fu**[4,*], **Jianguo Liu**[1,2,3], **Wenqing Liu**[1,2,3]

[1]Key Laboratory of Environmental Optics and Technology, Anhui Institute of Optics and Fine Mechanics, Chinese Academy of Sciences, Hefei, 230031, China

[2]University of Science and Technology of China, Hefei, 230027, China

[3]Center for Excellence in Regional Atmospheric Environment, Institute of Urban Environment, Chinese Academy of Sciences, Xiamen, 361021, China

[4]State Key Laboratory of Atmospheric Boundary Layer Physics and Atmospheric Chemistry, Institute of Atmospheric Physics, Chinese Academy of Sciences, Beijing, 100029, China

[5]University of Chinese Academy of Sciences, Beijing, 100049, China

[6]State Key Joint Laboratory of Environmental Simulation and Pollution Control, College of Environmental Sciences and Engineering, Peking University, Beijing, China

[*]now at: Institute of Surface-Earth System Science, Tianjing University, Tianjing, 300072, China

**Correspondence:** Min Qin (mqin@aiofm.ac.cn)

**Abstract.** Nitrous acid (HONO), an important precursor of the hydroxyl radical (OH), plays a key role in atmospheric chemistry, but its sources are still debated. The production of HONO on aerosol surface or on ground surface in nocturnal atmospheres remains controversial. The vertical profile provides vertical information on HONO and NO$_2$ to understand the nocturnal HONO production and loss. In this study, we report the first high-resolution (<2.5 m) nocturnal vertical profiles of HONO and NO$_2$ measured from in-suit instruments on a movable container that was lifted on the side wiring of a 325-m meteorological tower in Beijing, China. High-resolution vertical profiles revealed the negative gradients of HONO and NO$_2$ in nocturnal boundary layers, and a shallow inversion layer affected the vertical distribution of HONO. The vertical distribution of HONO was consistent with stratification and layering in the nocturnal urban atmosphere below 250 m. The increase of HONO/NO$_2$ ratio was observed throughout the column from the clean episode to the haze episode, and a relatively constant

HONO/NO$_2$ ratios in the residual layer were observed during the haze episode. Direct HONO
emissions from traffic contributed 29.3% ± 12.4% to the ambient HONO concentrations at night. The
ground surface dominates HONO production by heterogeneous uptake of NO$_2$ during clean episodes.
In contrast, the HONO production on aerosol surface (30–300 ppt) explained the observed HONO
increases (15–368 ppt) in the residual layer, suggesting that the aerosol surface dominates HONO
production aloft during haze episodes, while the surface production of HONO and direct emissions into
the overlying air are minor contributors. Average dry deposition rates of 0.74 ± 0.31 and 1.55 ± 0.32
ppb h$^{-1}$ were estimated during the clean and haze episodes, implying that significant quantities of
HONO could be deposited to the ground surface at night. Our results highlight ever-changing
contributions of aerosol and ground surfaces in nocturnal HONO production at different pollution
levels and encourage more vertical gradient observations to evaluate the contributions from varied
HONO sources.
**1 Introduction**

It is well known that the rapid photolysis of nitrous acid (HONO) (R1) after sunrise is the most

important hydroxyl radical (OH) source. 25%–60% of daytime OH production was accounted for due
to HONO photolysis, according to previously reported (Lu et al., 2012; Ma et al., 2017; Tong et al.,
2016; Su et al., 2008b; Huang et al., 2017; Spataro et al., 2013). OH initiates daytime photochemistry
and promotes the formation of secondary products (including ozone (O$_3$) and peroxyacetyl nitrate
(PAN)) and secondary aerosols (Alicke and Platt, 2002; Tang et al., 2015; Kleffmann, 2007; An et al.,
2012). In addition, HONO as a nitrosating agent forms carcinogenic nitrosamines (Hanst et al., 1977;
Pitts et al., 1978), and its health effects have attracted increasing amounts of concern (Sleiman et al.,
2010; Bartolomei et al., 2015; Gómez Alvarez et al., 2014).

$HONO + h\nu (320nm < \lambda < 405nm) \rightarrow NO + OH$                    (R1)

Despite the importance of HONO, the details of the formation processes of HONO in the

atmosphere are debated for decades. New state-of-the-art science instruments have observed much
higher daytime HONO concentrations than simulated values from atmospheric chemical models in both
rural and urban areas, implying missing HONO sources (Li et al., 2012; Wang et al., 2017; Oswald et
al., 2015; Wong et al., 2012; Li et al., 2014; Liu et al., 2019; Karamchandani et al., 2015; Kleffmann,
2007; Mendez et al., 2017; Michoud et al., 2014; Michoud et al., 2015; Tang et al., 2015; Vogela et al.,
2003; Sörgel et al., 2011). Several homogeneous reaction mechanisms for HONO have been proposed,
but the latter have been considered as irrelevant under actual atmospheric conditions, including
photolysis of ortho-substituted nitroaromatics (Bejan et al., 2006) and the reaction of photoexcited $NO_2$
with $H_2O$ (Li et al., 2008). The heterogeneous reduction of $NO_2$ with organic substrates is proposed to
be another effective pathway to generate HONO (Brigante et al., 2008; Stemmler et al., 2006; George
et al., 2005). However, extrapolation of lab results to real surfaces remains challenging. The nocturnal
production of HONO has been considered to be dominated by the $NO_2$ heterogeneous reaction (R2).
Although the heterogeneous reaction (R2) of HONO formation is first-order in $NO_2$, the mechanism for
the conversion of $NO_2$ on surfaces remains unclear (Finlayson-Pitts et al., 2003; Finlayson-Pitts, 2009).
$$2NO_{2(g)}+H_2O_{(ads)} \xrightarrow{surface} HONO_{(g)}+HNO_{3(ads)} \tag{R2}$$

A few studies have evaluated the relative importance of aerosol and ground surfaces in the

nocturnal production of HONO via reaction (R2). The heterogeneous reaction on ground surface have
been suggested as the primary nocturnal HONO source based on vertical measurements and fluxes in
HONO (Harrison and Kitto, 1994; Harrison et al., 1996; Laufs et al., 2017; Kleffmann et al., 2003; Su
et al., 2008b; VandenBoer et al., 2013; Villena et al., 2011; Wong et al., 2011; Wong et al., 2013; Stutz
et al., 2002; Ye et al., 2017; Zhang et al., 2009). However, other ground level studies have found
significantly positive correlations between $HONO/NO_2$ and aerosol surface areas, which suggests that
the aerosols play an important role in the heterogeneous conversion of $NO_2$ to HONO (Reisinger, 2000;
Cui et al., 2018; Zhang et al., 2018; Hou et al., 2016; Tong et al., 2016; An et al., 2012; Bao et al., 2018;
Liu et al., 2014; Reisinger, 2000). Therefore, the primary reaction surfaces for the nighttime HONO
formation is still controversial, and the role of the aerosols in the heterogeneous production of HONO
remains an open question.

Vertical gradient observations provide evidence regarding surfaces and in situ HONO formation,

which can help to understand the nighttime HONO sources. Methods of long-path differential optical
absorption spectroscopy (LP-DOAS) (Stutz et al., 2002; Wong et al., 2011; Wong et al., 2012),
instruments mounted on a movable elevator of a tall tower or a fixed height on a building (Kleffmann
et al., 2003; VandenBoer et al., 2013; Villena et al., 2011) and aircraft measurements (Zhang et al.,
2009; Li et al., 2014; Ye et al., 2018) have been applied for HONO vertical gradient observations in
Europe and the Americas. To determine the surface responsible for nocturnal HONO formation,
Kleffmann et al. (2003) and Wong et al. (2011) measured the HONO vertical gradient between 10 and
190 m altitude in a semi-rural area in Germany and at three different height intervals (lower: 20–70 m,
middle: 70–130 m and upper: 130–300 m) in downtown Houston. Their consistent conclusion was that
the reaction on the ground surfaces dominated the nocturnal formation of HONO. However, these types
of measurements are limited by the measurement frequency or vertical resolution between the surface
and the planetary boundary layer (PBL). VandenBoer et al. (2013) performed measurements of high
resolution vertical profiles (vertical resolution ~10 m) of HONO on a 300-m tower. The total column
observations of HONO also showed the ground as the dominant nocturnal surface on which HONO
was generated from the heterogeneous reaction of $NO_2$. The vertical information of HONO were
interpreted in conjunction with a chemical model. The results suggested a conservative surface
reservoir that was formed by the deposition of HONO could be a significant fraction of the unknown
daytime source. Furthermore, in an attempt to understand the importance of HONO photochemistry in
the troposphere, HONO gradients were measured in the PBL and the lower free troposphere (FT) over
a forested region in Michigan (Zhang et al., 2009). An evaluation of the relative importance of aerosol
and ground surfaces for the heterogeneous production of HONO also suggested that the ground surface
was a major HONO source in the lower boundary layer. In addition, a substantial amount of daytime
HONO existed in the FT (~8 ppt).
Beijing, as the largest and the most densely populated city in China, has suffered from severe haze
pollution for several years due to rapid economic development and urbanization. Several ground-based
observations of HONO have been conducted in urban and suburban areas of Beijing in recent years
(Tong et al., 2016; Zhang et al., 2018; Hou et al., 2016; Wang et al., 2017; Lu et al., 2012; Hendrick et
al., 2014). Higher levels of HONO have been observed (up to 9.71 ppb) in Beijing during winter
(Spataro et al., 2013). Although few near real-time HONO vertical gradients have been made, and they
have suggested that the reaction at the ground surface is the most important nighttime HONO source
(Kleffmann et al., 2003; VandenBoer et al., 2013; Wong et al., 2012; Zhang et al., 2009), the relative
importance of aerosol and ground surface in the production of nocturnal HONO may be different in the
Beijing region. First, as the primary precursor of nighttime HONO, $NO_2$ has a much higher
concentration during winter in Beijing due to the burning of fossil fuels and vehicle emissions. Second,
the aerosol surface area has been reported to be two to three orders of magnitude higher than the typical
background area (Cai et al., 2017; Liu et al., 2012; Zhang et al., 2015). High aerosol surface area levels
favor aerosol surfaces to play a heterogeneous reaction surface (haze period: 3000 $\mu m^2$ $cm^{-3}$; Wang et
al., 2018), which presumably makes aerosol surfaces to play a more important role in the production of
nighttime HONO. Third, there is more stable nocturnal stratification during the haze period in winter in
Beijing, which may have influenced the vertical distribution of HONO. The contribution of the surface
production of HONO to HONO levels aloft may be overestimated.
In this study, the first high-resolution vertical profile measurements of HONO and $NO_2$ in the
megacity of Beijing at different pollution levels (following the transition from a clean episode to a haze
episode) are reported. The vertical profiles of HONO and $NO_2$ are measured at high vertical resolution
(< 2.5 m over 240 m height) between the surface, the nocturnal boundary layer, and the residual layer.
Although the vertical profile measurements are rather limited in scope, including only four nights in
December 2016, with limited ancillary data, this study is unique due to the high vertical resolution
obtained and due to the continuous HONO and $NO_2$ vertical measurements obtained at different stages
of pollution. The vertical profiles are then interpreted to evaluate the aerosol and ground surfaces
responsible for the nighttime HONO formation during different pollution periods. The vertical
measurements and simultaneous observations at ground level are then used to identify and quantify
nighttime HONO sources.
**2 Experimental Methods**
**2.1 Measurement site**
Vertical profile measurements were conducted from December 7th to 12th of 2016 at the Tower
Branch of the Institute of Atmospheric Physics (IAP), Chinese Academy of Science (39°58'N,
116°23'E) as part of the "In-depth study of air pollution sources and processes within Beijing and its
surrounding region (APHH-Beijing)" winter campaign. The site is a typical urban residential area
located between the 3rd and 4th Ring Road in the north of Beijing. It is approximately 1 km from the 3rd
Ring Road, 200 m from the Beijing-Tibet Expressway, and 50 m from the Beitucheng West Road (Fig.
S1). The primary sampling platform was the Beijing 325-m meteorological tower (BMT), equipped
with an external container that was lifted on the side wiring of the tower, which could ascend and
descend at a relatively constant rate of ~ 9 m $min^{-1}$. A single vertical ascent or descent required less
than 30 min. After reaching the top, the container stopped and data were measured continuously for 5–
20 min of each cycle. For security reasons, the container reached a maximum height limit of 260 m
during the daytime and 240 m at night (Fig. 1). The container instruments included the following: a
global position system (GPS), an altimeter, and an incoherent broadband cavity enhanced absorption
spectrometer (IBBCEAS) for measurements of HONO and $NO_2$. In addition, another IBBCEAS was
mounted in temperature-stabilized lab containers for the measurement of HONO and $NO_2$ at ground
level.
**2.2 Instrumentation**

HONO and $NO_2$ were simultaneously measured using a home-made IBBCEAS. A detailed

description of the IBBCEAS instrument can refer to Duan et al (2018), and its application to the
measurement made during this study is described below. IBBCEAS is a spectroscopic technique that
combines broad-band light source (UV-LED) with the principle of time-integrated cavity output
spectroscopy. The HONO was sampled into an inlet tube (1.5 m length with a 4 mm outside diameter
(OD)) before entering an optical cavity (550 mm in length and 25.4 mm OD) that utilized PFA to
minimize the HONO loss. The sampling gas flow rate was controlled at six standard liters per minute
(SLPM) by a gas pump (KNF). In the optical cavity, light was reflected between the two highly
reflective mirrors (R = 99.980% @368 nm, CRD Optics, California, USA) to obtain a long optical
absorption length (the total optical path ~4.5 km). To protect the highly reflective mirrors, pure $N_2$ was
used to continuously purge the mirrors to prevent contact between the mirrors and the sample airflow.
The purge flow rate was controlled at 0.1 SLPM using mass flow controllers (MFCs, CS200A,
Sevenstar, Beijing, China). In this study, the IBBCEAS instrument was mounted in a movable container
of the BMT for vertical profile measurements, and this made measurements with a time resolution of
15 s (vertical resolution of 2.4 m). The 3 σ detection limits for HONO and $NO_2$ were 360 ppt and 600
ppt, respectively. Another IBBCEAS instrument was mounted in temperature-stabilized lab containers
at ground level, and it collected data with a time resolution of 30 s. The detection limits for HONO and
$NO_2$ were 270 ppt and 510 ppt, respectively. The total relative uncertainty of the IBBCEAS instrument
was 8.7%, and it considered the uncertainty in the cross section (5%), the calibration of reflectivity
(5%), spectral fitting (4%), the effective cavity length (3%), the pressure in the cavity (1%), $\Delta I/I_0$
(1%), and sample loss (0.5%). Correction of the light intensity was performed every hour, and the
mirror reflectivity was calibrated daily.

Meteorological parameters that included wind speed (WS), wind direction (WD), temperature (*T*),

and relative humidity (RH) were obtained using a 15-level meteorological gradient observation system
installed at fixed intervals along the meteorological tower (at heights of 8, 15, 32, 47, 65, 80, 100, 120,
140, 160, 180, 200, 240, 280, and 320 m). The gaseous species, including nitrogen monoxide (NO),
ozone ($O_3$), and carbon monoxide (CO) were measured using a commercial gas analyzer from Thermo
Scientific (Waltham, Massachusetts, USA) (Tan et al., 2017). NO was detected using NO-$O_3$
chemiluminescence (Model 42iTL, Thermo Scientific), with an accuracy of ±20% and a detection limit
of 50 ppt. $O_3$ and CO were measured by an $O_3$ analyzer (Model 49i, Thermo Scientific) and a CO
analyzer (Model 48iTLE, Thermo Scientific), with the detection limits of 0.50 ppb and 0.04 ppm, and
an $O_3$ accuracy of ±20%. The 7-wavelength aethalometer (AE33, Magee Scientific Corp, Berkeley,
California, USA) was deployed to measure the black carbon (BC) at a time resolution of 1 min (Xie et
al., 2019). Aerosol particles were continuously collected onto a quartz filter in the instrument to
measure their light attenuation at 370, 470, 520, 590, 660, 880, and 950 nm. Trace gas (CO and $O_3$) and
aerosol parameters (BC, NR-$PM_1$ and aerosol surface area) were measured simultaneously at ground
level and at 260 m on the tower, while NO was measured only at ground level. The non-refractory
submicron aerosol (NR-$PM_1$) species were measured simultaneously at ground level and at 260 m
using an Aerodyne high-resolution time-of-flight aerosol mass spectrometer (AMS) and an aerosol
chemical speciation monitor (ACSM), respectively. The detailed sampling setup and calibration of the
AMS and ACMS, as well as data analysis, have been described by Xu et al. (2019) and Sun et al.
(2013). The dry-state particle number size distributions were measured at ground level and at 260 m
using a scanning mobility particle sizer (SMPS) (Du et al., 2017). The particle number size
distributions of 15-500 nm was used to calculate the aerosol surface area ($S_a$) by assuming the particles
are in spherical shape. A hygroscopic factor $f$(RH) was applied to correct $S_a$ to the aerosol surface area
in the real atmosphere ($S_{aw}$) (Li et al., 2012). The $S_{aw}$ was calculated using following equations:
$$f(RH) = 1 + a\left(\frac{RH}{100}\right)^b \tag{1}$$

$$S_{aw} = S_a \times f(RH) \tag{2}$$

where $a$ and $b$ are the empirical fitting parameters used to estimate $f$(RH), which were set to 2.06 and
3.6 in urban region (Liu et al., 2008). The curve-fitting parameters $a$ and $b$ were derived from the
measurements in Guangzhou region, which, like Beijing, is one of the mega-cities in China. The
uncertainty of $S_{aw}$ was estimated to be ~30%, which was associated with the uncertainty from the $S_a$
measurement (~20%) and the growth factor (~20%).
**2.3 Inter-comparison**

In the present study, the measurements of HONO and $NO_2$ were conducted simultaneously in the

container and at ground level. Therefore, the calibration and inter-comparison of the two IBBCEAS
instruments were crucial. Comparison experiments were conducted in a temperature-stabilized
laboratory. The sampling unit and sampling flow rate of the two instruments were identical to minimize
measured deviations. Figure. 2 shows the excellent agreement between the two IBBCEAS instruments
(HONO: $R^2 = 0.99$, $NO_2$: $R^2 = 0.99$), with a slope of $1.00 \pm 0.01$ ($NO_2$), $1.00 \pm 0.01$ (HONO) and a
small intercept of $180 \pm 90$ ppt ($NO_2$) and $-10 \pm 10$ ppt (HONO).

To verify the relative accuracy of the IBBCEAS instrument, an inter-comparison between the

IBBCEAS of this study and the IBBCEAS of Cambridge University was conducted. The HONO
measurements from the two different IBBCEAS instruments were highly consistent ($R^2 = 0.98$, Fig. 2c),
with a small intercept and a slope close to 1. The difference of 4% was within the range of the
instrumental measurement error. In addition, the IBBCEAS instrument was also compared with the
long optical path absorption photometer (LOPAP) and the stripping coil ion chromatography (SC-IC)
from our previous studies (Tang et al., 2019; Duan et al., 2018). This also showed good agreements of
the HONO measurements (LOPAP: $R^2 = 0.894$, SC-IC: $R^2 = 0.98$). The regression slope and intercept
of LOPAP to IBBCEAS was $0.941 \pm 0.0069$ and $0.110 \pm 0.0089$, respectively, with a difference of ~6%.
The linear regression of IBBCEAS against the SC-IC exhibited a slope of 0.82 with an intercept of 0.22.
The difference of ~8% between the two instruments may be caused by the gas sampling.
**3 Results and discussion**
**3.1 General observations and vertical measurements**

The time-series of meteorological parameters, trace gases, and aerosol parameters are shown in

Fig. 3. Based on the NR-PM$_1$ mass concentration level, three different meteorological conditions were
characterized during the measurement periods (Table 1). The first episode (E1) from December 7[th] to
10:00 on December 8[th] was a haze event. The NR-PM$_1$ mass concentration increased rapidly from 30 to
~150 $\mu g \cdot m^{-3}$ at ground level and at 260 m on the tower due to a low wind speed ($0.78 \pm 0.42$ m·s$^{-1}$) and
a high RH (51% $\pm$ 13%).

The second episode (December 8-11, C2) was a clean event with low NR-PM$_1$ mass loading

(mean: $24 \pm 19$ $\mu g \cdot m^{-3}$) and a high wind speed ($> 5$ $m \cdot s^{-1}$), primarily from northwest. The third episode
(E3) from December 11th to December 12th was another haze event. During this period, the atmosphere
was characterized by stagnant weather, lower wind speeds (an average of $0.77 \pm 0.4$ $m \cdot s^{-1}$) and a high
RH ($55\% \pm 5\%$). The mass concentration of the NR-PM$_1$ gradually increased and then remained at
relatively constant levels at ground level and 260 m on the tower, and ranging from 69 to 218 $\mu g \cdot m^{-3}$
with a mean value of $154 \pm 35$ $\mu g \cdot m^{-3}$.

Throughout the entire measurement periods, HONO concentrations ranged from 0.27 to 7.59 ppb.

The mean HONO mixing ratios during E1, C2, and E3 were $4.26 \pm 2.08$, $0.90 \pm 0.65$, and $3.54 \pm 0.91$
ppb, respectively. The maximum HONO concentration was 7.59 ppb, which was observed during E1
(at 08:10 on December 8th). From December 11th to 12th the pollutants continuously increased with the
stagnant weather. The HONO concentrations remained at high levels, and the daytime mean HONO
mixing ratio even reached $3.10 \pm 0.92$ ppb. Figure 3 also shows the time series of simultaneously
measured relevant species. The mean NO$_2$ mixing ratios during E1, C2, and E3 were $51.98 \pm 8.41$,
$23.30 \pm 11.91$, and $51.88 \pm 5.97$ ppb, respectively. Because NO and O$_3$ were not measured at ground
level after 14:00 on December 10th, the mean concentrations of NO and O$_3$ during E1 were $90.99 \pm$
$67.98$ and $14.66 \pm 21.79$ ppb, while the concentration of NO and O$_3$ during C2 were $4.04 \pm 1.81$ and
$14.37 \pm 10.65$ ppb, respectively. After sunset, the concentration of O$_3$ at the surface was rapidly titrated
due to the elevated NO and increased with an increase in height. The mixing ratio of O$_3$ below 260 m
was less than 9 ppb during the vertical measurements. The BC, NR-PM$_1$, and aerosol surface area
showed very similar patterns both at ground level and at 260 m. The RH corrected aerosol surface area
($S_{aw}$) is shown in Fig. S2. Higher BC, NR-PM$_1$ and $S_{aw}$ levels were observed at ground level during the
haze periods (E1 and E3).

Nocturnal stable surface layers of air generally form at low wind speeds ($< 6$ $m \cdot s^{-1}$) (VandenBoer

et al., 2013). Hence, the vertical profile data were used when the wind speeds were less than 6 $m \cdot s^{-1}$,
except on December 7th. Vertical measurements during low wind events were successfully conducted
on three occasions (December 9–10, 10–11, and 11–12) and will be discussed below. The
near-continuous vertical measurements avoided the observation bias from prolonged fixed sampling.
The date and time of the measurement for each vertical profile is detailed in Table S1 in the
supplementary information.

**3.2 Nocturnal HONO vertical profiles**

**3.2.1 Vertical measurements after sunset**

Vertical measurements were conducted from ground level to 240 m after sunset. The mixing ratios of HONO and $NO_2$ at ground level were consistent with those measured in the container. The mixing ratios of HONO and $NO_2$ showed nearly flat profiles throughout the column during C2 and E3 (Fig. S3), indicating that HONO and $NO_2$ were relatively well mixed after sunset The vertical variations of △HONO, which is the difference in the HONO concentrations between measured in the container and at ground level, centered around 0 ppb. The variation of △HONO throughout the column were close to the detection limit of the IBBCEAS instrument and were barely observed. This result also indicated the relatively uniform vertical distribution of HONO. The vertical variations in $T$ and RH during these three vertical measurements were similar (Fig. S4). While $T$ decreased gradually with increasing height, RH increased gradually with increasing height. In addition, RH was relatively higher during the haze episode. Also, there was no $T$ inversion just after sunset, and the consistent variations in the HONO and $NO_2$ at ground level and in the vertical measurements all supported a relatively well-mixed boundary layer, which explained the uniform vertical distribution of HONO and $NO_2$.

**3.2.2 Nocturnal vertical profiles**

Nocturnal small-scale stratification and layering was determined according to the method of Brown et al. (2012), who used the potential temperature profile as an indicator of atmospheric static stability. According to the vertical variations in the potential temperature, the stable layer was divided into the "surface layer", the "nocturnal boundary layer (NBL)", the "top of the nocturnal boundary layer", and the "residual layer (RL)".

Figure 4 depicts the nocturnal vertical profiles of HONO, $NO_2$, and potential temperature during C2. The linear least squares regression slope and correlation coefficients of HONO and $NO_2$ to altitude were applied to estimate the nocturnal gradient of HONO and $NO_2$ (Table 2). Vertical profile data were used above the surface layer and 10 m vertical average from the surface to 240 m AGL to evaluate vertical gradient of HONO and $NO_2$. An example of the regression line was shown in Fig. S5. On the night of December 9[th] (Fig. 4a), negative profiles of both HONO and $NO_2$ were clearly seen. When the container ascended during 22:42–23:06, the potential temperature profile showed distinct stratification. The surface layer extended to 10–20 m and the NBL extended to ~140 m. The obviously negative gradient of HONO (-4.56 $\pm$ 0.34 ppt m$^{-1}$) and $NO_2$ (-16.41 $\pm$ 1.22 ppt m$^{-1}$) were observed throughout

the heights from 0 to 240 m. A negative gradient of HONO was observed in the RL, but was not
consistently observed in other measurements (see below). During the descent of the container from
23:15–23:40, the potential temperature profile showed that a shallow $T$ inversion had rapidly formed
between 130 and 200 m. The obvious vertical variation in RH during 23:15–23:40 (Fig. S3) also
indicated the different layers at different heights, which was due to the influence of a shallow inversion
layer. Within the shallow inversion layer, vertical convection and transport were inhibited, and a
remarkable negative gradient was observed there. However, within the NBL, the negative gradient of
HONO and $NO_2$ disappeared. This might have been due to the continuous vertical mixing below the
shallow inversion layer from 23:06 to 23:40. Additionally, the surface source of HONO was obvious, as
evidenced by the apparently negative gradient of HONO in Table 2.

The vertical profile of potential temperature on December 10[th] (Fig. 4b) showed that a shallow

inversion layer formed between the surface layer and the NBL. In the shallow inversion layer, the
mixing ratios of HONO decreased rapidly with increasing height, and a significant negative gradient
was observed within the shallow inversion layer and surface layer. With the attenuation of the shallow
inversion layer during the descent of the container from 23:01 to 23:25, the inhibition of vertical
transport and mixing gradually weakened. The increase in the negative gradient of HONO and $NO_2$ and
the correlation coefficients of HONO and $NO_2$ to altitude from 22:36 to 23:25 also showed the
weakened shallow inversion layer near the surface, which suggested the nighttime HONO surface
source. The attenuation event of the shallow inversion layer may have also been the result of an
increase in the wind speed and the interaction of different air masses that changed from the west to
southeast between 15 and 100 m (Fig. S6). Above the 100 m height, the mixing ratio of HONO
decreased with increasing height, and the fluctuation in HONO was likely due to the interaction of
different air masses. In contrast, the vertical profiles of $NO_2$ showed that $NO_2$ rapidly decreased
towards the ground, and a significant positive gradient was observed near the surface that was caused
by several factors. Nocturnal $NO_2$ is produced by the reaction of $O_3$ with NO, which primarily occurs
near the surface, resulting in a negative gradient in $NO_2$. However, this effect was counteracted by the
dry deposition of $NO_2$, which by itself would result in a positive gradient (Stutz et al., 2004b).
Additionally, the mixing ratio of $NO_2$ was also affected by local traffic emission sources, and a
near-surface shallow inversion layer was formed on December 10[th]. All of these presumably led to a
clearly positive gradient for the near-surface $NO_2$. In contrast to the vertical profiles measured on
December 9$^{th}$, a positive gradient observed in near-surface $NO_2$ on December 10$^{th}$ indicated that the
shallow inversion layer affected the vertical distribution of HONO and $NO_2$ at night.
Although the surface layer was a common feature in the potential temperature profiles, it was
absent during E3, and the NBL extended downward to the lowest measurement height (8 m above the
ground). As shown in Fig. 5, the vertical profile of HONO showed a significant negative gradient as the
container ascended during 22:35–23:00, and higher HONO mixing ratio was observed at ground level.
With the development of the boundary layer, the negative gradient of HONO continued to decrease
from $6.92 \pm 0.36$ ppt m$^{-1}$ during 22:35–23:00 to $1.98 \pm 0.28$ ppt m$^{-1}$ during 00:45–01:09 and even
disappeared between 00:00 and 00:26. Moreover, the consistent HONO/$NO_2$ ratios (~5.6% $\pm$ 0.3%)
were observed throughout the depth of the NBL between 23 and 01 h (Fig. S7). A near-steady state
plateau of the HONO mixing ratio and HONO/$NO_2$ was established near midnight with the NBL.
Similar vertical measurements were reported by VandenBoer et al (2013), who also observed a
near-steady state in the HONO mixing ratio and HONO/$NO_2$, and an approximate balance between the
production and loss of HONO late in the night. A possible physical and chemical process, the loss of
HONO to the ground surface due to dry deposition could account for the buildup and near-steady state
observed in the HONO mixing ratio and HONO/$NO_2$. This implied that significant quantities of HONO
were deposited to the ground surface at night.
The utility of the linear least squares regression slope of HONO to altitude to estimate the vertical
gradient of HONO at night implies that the potential nocturnal HONO production from the
heterogeneous reaction of $NO_2$ on aerosol surface. A positive gradient of HONO ($0.24 \pm 0.39$) between
00:00 and 00:26 was observed during E3. The aerosol surface area ($S_{aw}$) in the residual layer was
greater than 1500 µm$^2$ cm$^{-3}$ throughout the night (range: 1592-2655 µm$^2$ cm$^{-3}$). The $S_{aw}$ was 2314 µm$^2$
cm$^{-3}$ from 22 to 01 h on December 11$^{th}$ and reached a maximum of 2569 µm$^2$ cm$^{-3}$ in the residual layer.
These aerosol surface areas are a factor of 14-38 greater than that observed in previous studies of
HONO vertical gradient, which ranged between 60 and 158 µm$^2$ cm$^{-3}$ (Kleffmann et al., 2003;
VandenBoer et al., 2013). Such high aerosol surface areas may provide a sufficient surface for the
heterogeneous reaction. The vertical profiles also showed an enhanced HONO/$NO_2$ ratios from C2 to
E3 (Fig. S8). Moreover, a relatively constant HONO mixing ratio and HONO/$NO_2$ ratio above 160 m
were observed from 22:35 to 01:09 during E3. Both of these observations are indicative of a potential
aerosol surface source of HONO aloft during the haze episode. Assuming that aerosol surface
production dominated the observed HONO mixing ratio in the overlying air during the haze episode,
the mixing ratios of HONO and $NO_2$ observed at 240 m and the aerosol surface area measured at 260 m
were parameterized to estimate the nocturnal production of HONO on aerosol surface, which is
explored in more detail in section 3.4.2.
**3.3 Direct emissions**
In the present study, the measurement site was surrounded by several main roads, and thus might
have been affected by vehicle emissions. CO and NO, as the primary pollutants, are emitted from
combustion processes like the burning of fossil fuels as well as vehicle emissions (Sun et al., 2014;
Tong et al., 2016; Bond et al., 2013). BC is another primary pollutant typically emitted from diesel
engines and residential solid fuels (Zhang et al., 2018). Good correlations of the nocturnal HONO with
CO ($R^2$=0.85), NO ($R^2$=0.76) and BC ($R^2$=0.84) at ground level were observed (Fig. S9), indicating the
potential effect of direct emissions on the observed HONO at night. The emission ratio of $HONO/NO_x$
have been determined from tunnel measurements in California (Kirchstetter et al., 1996), Germany
(Kurtenbach et al., 2001), and Hong Kong (Laing et al., 2017). However, considering the differences in
the type of vehicles, fuel compositions, and other factors, the reported emission factor of $HONO/NO_x$
might not be representative for the Beijing region. To evaluate the influence of direct emissions, the
local emission factor of HONO was derived from ambient measurements. Since NO was not measured
at ground level after December 10[th], the nighttime measurement data of HONO and $NO_x$ from
November 9[th] to December 10[th] were used to evaluate the local HONO emission factor.
Considering the potential secondary HONO formation with air mass aging during the transport
process, five criteria were applied to ensure as much of the freshly emitted air masses as possible: (a)
only nighttime data (from 18:00 LT to next 6:00 LT) were included to avoid the fast photolysis of
HONO; (b) only sharp peaks during nighttime and the elevations of HONO and $NO_x$ over the
background levels were estimated; (c) $\Delta NO/\Delta NO_x > 0.80$; (d) good correlation between HONO and
$NO_x$; (e) short duration of the plume (< 30 min). The typical nighttime wind speed at measurement site
was 1.2 m s$^{-1}$, thus the duration for fresh air masses should have been less than 30 min during transport
from the emission to the measurement site. Criteria (b) and (c) were used as indicators for identifying
fresh vehicular emissions. Criteria (d) and (e) further confirmed that the increase in HONO was
primarily caused by freshly emitted plumes instead of heterogeneous reactions of $NO_2$.
Figure 6 shows two emission plumes observed on December 9th to 10th, 2016 based on the
preceding selection criteria. The slopes of HONO to $NO_x$ can be considered as the emission ratios
(Rappenglück et al., 2013). The HONO/$NO_x$ emission ratios were estimated for the 11 fresh emission
plumes that satisfied the preceding criteria (see Table 3). The derived emission factors of 0.78%–1.73%
had an average value of 1.28% ± 0.36%, which was larger than the 0.53%–0.8% measured in the tunnel
in Wuppertal (Kurtenbach et al., 2001). The minimum ratio of 0.78% approximated the value (0.8%)
measured in Wuppertal. It is worth mentioning that the value of 0.8% is widely used as the upper limit
of the HONO/$NO_x$ emission ratio from road traffic in interpreting field observations and modeling
HONO emissions (Stutz et al., 2002; Su et al., 2008a; Tong et al., 2016). The maximum ratio of 1.73%
in this study is comparable to the value of 1.7% in Houston, Texas, observed by Rappenglück et al.
(2013). The derived emission ratios were within the range of other published results (0.19%–2.1%)
(Kirchstetter et al., 1996; Kurtenbach et al., 2001; Su et al., 2008a; Rappengluck et al., 2013; Yang et
al., 2014; Xu et al., 2015; Liang et al., 2017; Zhang et al., 2018; Li et al., 2018; Liu et al., 2019).
Comparisons of the derived HONO/$NO_x$ ratios with the results obtained previously are summarized in
Table S2. To minimize the risk of overestimating the direct emissions, the minimum HONO/$NO_x$ ratio
was used as an upper limit for the emission factor (Su et al., 2008a). The minimum HONO/$NO_x$ ratio of
0.78% was used to evaluate the contribution of vehicle emissions to the ambient HONO levels at night
(Eq. (3)). In this case, the risk of overestimating direct emissions was minimized, but there was still the
effect of potential secondary HONO formation.
$$[HONO]_{emis} = 0.0078 \times [NO_x] \tag{3}$$

where $[HONO]_{emis}$ and $[NO_x]$ are the HONO mixing ratios from vehicle emissions and the observed
$NO_x$ mixing ratios, respectively. The direct emissions contributed an average of 29.3% ± 12.4% to the
ambient HONO concentrations at night, with an average HONO$_{emis}$/HONO value of 35.9% ± 11.8%
during the clean episode and an average HONO$_{emis}$/HONO value of 26% ± 11.3% during the haze
episode. The frequency distribution of HONO$_{emis}$/HONO during the clean and the haze episodes are
shown in Fig. 7. The lower vehicle emissions contribution during the haze episode could have been
caused by an odd-even car ban, which required alternate driving days for cars with even- and
odd-numbered license plates.
**3.4 Nocturnal HONO chemistry**
**3.4.1 Correlation studies**
The heterogeneous conversion of $NO_2$ is an important pathway for HONO formation during the
nighttime, as many field observations have found a good correlation between HONO and $NO_2$ (Zhou et
al., 2006; Su et al., 2008a; Wang et al., 2013; Huang et al., 2017). However, the use of a correlation
analysis to interpret the heterogeneous conversion of $NO_2$ should be treated carefully, as physical
transport and source emissions also contribute to the correlation. In this study, the correlations of
vertical profiles between HONO and $NO_2$ were analyzed. Vertical profile data without horizontal
transport were used to avoid the influence of physical transport. As shown in Fig. 8, the linear least
squares regression correlations of HONO to $NO_2$ exhibited moderate but significant correlations (C2:
$R^2 = 0.72$, E3: $R^2 = 0.69$), supporting that $NO_2$ participated in the formation of HONO. The column of
HONO and $NO_2$ showed a significantly positive correlation during the haze episode. However, the
negative correlation between HONO and $NO_2$ was observed at ground level during the haze episode
(Fig. S10), which was also observed in the previous ground-based observations (Hou et al., 2016;
Zhang et al., 2018). The observed significant correlation between the HONO column and $NO_2$ column
could be due to: (1) emissions and vertical mixing of HONO and $NO_2$, and (2) a possible
heterogeneous reaction of $NO_2$ on aerosol surface.
Adsorbed water on a surface has been shown to affect the heterogeneous formation of HONO
(Stutz et al., 2004a). The relationship between HONO/$NO_2$ and RH is illustrated in Fig. 9. Following
the method introduced by Stutz et al (2004a), the average of the five highest HONO/$NO_2$ values in each
10% RH interval was evaluated to eliminate much of the influence of factors like the time of night,
advection, the surface density, etc. An increase in the HONO/$NO_2$ ratio along with RH was observed at
each height interval when the RH was less than 70%. A previous observation at ground level also
reported that the HONO/$NO_2$ ratio increased with an increase in RH when the RH was less than 70%. A
further increase in RH would lead to a decrease in the HONO/$NO_2$ ratio, which was considered to be
caused by the number of water monolayers that formed on the surface leading to an efficient uptake of
HONO (Li et al., 2012; Yu et al., 2009; Liu et al., 2019). However, a decreased uptake coefficient of
HONO with increasing RH from 0% to 80% was observed in a laboratory study (Donaldson et al.,
2014). The $NO_2$ to HONO conversion efficiency depended negatively on RH at an RH above 70%,
which was presumably caused by the relative humidity affecting both HONO uptake onto the surface
and the $NO_2$-to-HONO conversion. A decrease in the $HONO/NO_2$ ratio with an increase in height at a
similar RH level were observed during C2 and E3. A higher conversion efficiency of $NO_2$ to HONO
was observed near the surface, and the $HONO/NO_2$ ratios at different heights were significantly
different during C2. However, this differences decreased, and similar $HONO/NO_2$ ratios were observed
at different heights during E3. This observation implied a possible heterogeneous conversion of $NO_2$ on
aerosol surface in the overlying air. It is necessary to note that the limited vertical measurements
resulted in a limited variation range in the RH, which limits this analysis. Additional efforts are needed
to conduct more comprehensive vertical measurements to interpret the $HONO/NO_2$ ratios versus RH
for different heights in the future.
**3.4.2 Relative importance of aerosol and ground surfaces in nocturnal HONO production**
The observed positive HONO gradient implied a potential heterogeneous conversion of $NO_2$ on
aerosol surface. The aerosol surface area observed during the haze episode was an order of magnitude
higher than in other studies of HONO vertical gradient (Kleffmann et al., 2003; VandenBoer et al.,
2013), which presumably provided sufficient aerosol surface area to account for the observed nighttime
HONO production (Liu et al., 2019). The surface area information for particles larger than 0.5 µm were
not valid at ground level and 260 m during the measurement periods. Hence, this is a lower limit
estimate of the total surface area for the heterogeneous reaction.
An estimate of the nocturnal HONO production on aerosol surface was made using the RH
corrected aerosol surface area ($S_{aw}$) and $NO_2$ observations from the residual layer. The CO and BC
measured at ground level were independent of the CO and BC observed at 260 m during the haze
period (Fig. S11), since it can be expected that air masses in the residual layer were decoupled from the
ground-level processes and largely free of $NO_2$ emissions. (Brown et al., 2012; VandenBoer et al.,
2013). The HONO production from the heterogeneous $NO_2$ conversion (Reaction R1) on aerosol
surface would then have become the primary HONO source in the residual layer during E3. The
reactive uptake of $NO_2$ by the aerosols was assumed to occur on all measured aerosol surface areas,
regardless of their chemical composition. HONO production (*P(HONO)*) can then be expressed using
the equation of Ye et al. (2018) modified to account for the disproportionation:
$$\frac{P(HONO)}{[NO_2]} = \frac{1}{8} \times S_{aw} \times \sqrt{\frac{8RT}{\pi M}} \times \gamma_{NO_2} \qquad (4)$$
where $\gamma_{NO_2}$ is the uptake coefficient, $R$ is the gas constant, $T$ is the absolute temperature (K), $M$ is the
molecular mass of $NO_2$ ($M = 4.6 \times 10^{-2}$ kg mol$^{-1}$), and $S_{aw}$ is the RH corrected aerosol surface area (µm$^2$
cm$^{-3}$). It was assumed that the uptake of $NO_2$ to form HONO would not cause a significant change in
$NO_2$ concentration over a time period. The $NO_2$-normalized HONO production over time, $\Delta \frac{[HONO]}{[NO_2]}/\Delta t$,
can be calculated using the following Eq. (5):

$$\Delta \frac{[HONO]}{[NO_2]}/\Delta t \sim \frac{1}{8} \times S_{aw} \times \sqrt{\frac{8RT}{\pi M}} \times \gamma_{NO_2} \qquad (5)$$

Assume an $NO_2$ uptake coefficient of $1 \times 10^{-5}$ to $1 \times 10^{-6}$ in the dark, which fits the $NO_2$ uptake
coefficient values observed in relevant studies (J.Kleffmanna et al., 1998; Kurtenbach et al., 2001;
Saastad et al., 1993; Bröske et al., 2003). A representative temperature of 273 K, and an average observed
$S_{aw}$ of 2314 µm$^2$ cm$^{-3}$ in the residual layer between 22 and 01 h during E3 were used. A relative HONO
accumulation rate of $\Delta \frac{[HONO]}{[NO_2]}/\Delta t$ ranged between 0.00037 and 0.0037 h$^{-1}$, equivalent to the HONO
production of 0.02 to 0.20 ppb h$^{-1}$ at a constant $NO_2$ concentration of 52.88 ppb, which was the average
value of the nocturnal $NO_2$ observed in the residual layer during E3. The absolute amount produced of
HONO in an interval of 1.5 h (30–300 ppt) could account for the observed increases of HONO (15–368
ppt) in the residual layer between vertical profile measurements on December 11$^{th}$ (time interval: 1.5 h).
Thus, production from the heterogeneous conversion of $NO_2$ on aerosol surface can explain the HONO
observations during E3. In addition, if the HONO production aloft was indeed dominated by reactions
on aerosol surface, the column average concentration of HONO would be expected to be independent
of the amount of HONO observed at ground level. Figure 10a shows that the column of HONO is
independent of the mixing ratio of HONO observed from the ground level to 10 m in height ($R^2 = 0.27$),
which is consistent with the hypothesis that the aerosol surface dominates HONO production aloft by
heterogeneous uptake of $NO_2$ during the haze episode, and the production of HONO at ground level
and direct HONO emissions into the overlying air are minor contributors. This result was contrary to
previous observations that the production of HONO on aerosol surface was insignificant compared to
the ground surface (Kleffmann et al., 2003; Wong et al., 2011; VandenBoer et al., 2013), which could
have been due to the higher aerosol surface area observed in this study. An order of magnitude higher
aerosol surface area in the residual layer than in previous vertical observations (<160 µm$^2$ cm$^{-3}$) was
observed, which could provide sufficient aerosol surface area for the heterogeneous formation of
HONO. However, the limited vertical profile dataset limits a comprehensive investigation of HONO
formation in Beijing, yet provides a data basis and research direction.
An estimate of HONO production from the heterogeneous conversion of $NO_2$ on aerosols was also
made during C2 using $S_{aw}$ and $NO_2$ observations from the residual layer. The column of the average
HONO concentration was related to the amount of HONO observed between ground level and 10 m
(Fig. 10b, $R^2 = 0.93$), suggesting that the surface HONO sources affected the HONO observed
throughout the depth of boundary layer during C2. A high correlation ($R^2 = 0.83$) between the measured
CO and BC at ground level and the CO and BC at 260 m was also observed (Fig. S10), which indicated
that vehicle emissions affected air masses in the residual layer. The lack of the NO vertical profile
cannot directly correct the influence of direct HONO emissions. If it is assumed that the contribution of
direct HONO emissions was consistent at ground level and in the residual layer, the relative
contribution of the aerosol and ground surfaces to nocturnal HONO production in the residual layer
could be roughly estimated during C2. The direct emissions contribution of 35.9% ± 11.8% during C2
is a higher limit estimate of the contribution of direct emissions to the HONO levels in the residual
layer.
The averages $S_{aw}$ of 791 and 894 $\mu m^2\ cm^{-3}$ from 17 to 24 h, and the average $NO_2$ mixing ratio of
34.67 and 42.40 ppb from the residual layer were used to estimate HONO production on aerosol
surface on December 9th and 10th. The formation rates of HONO on aerosol surface were 0.0045–0.045
ppb h$^{-1}$ on the 9th and 0.0059–0.059 ppb h$^{-1}$ on the 10th. The observed HONO increased by 305–608 ppt
between vertical profile measurements (time interval: 5.5 h), which have the contributions from direct
HONO emissions subtracted, were higher than the production of HONO (25–248 ppt) in an interval of
5.5 h on December 9th. The formation of HONO on aerosol surface cannot explain the observed HONO
increases in the residual layer, which suggests that the HONO observed in the residual layer was
primarily derived from the heterogeneous conversion of $NO_2$ on the ground surface followed by
vertical transport throughout the column. The aerosol production of HONO could account for up to
about 40% of HONO observations in the residual layer. The HONO production from the aerosol
surface in an interval of 5.35 h was 32–316 ppt on December 10th, which was comparable to the
corrected HONO increases of 114–369 ppt observed between vertical profile measurements (time
interval: 5.35 h). A shallow inversion layer formed near the surface could account for the aerosol
surfaces play a heterogeneous reaction surface in the residual layer.
In conclusion, HONO production solely on aerosol surface accounted for the HONO observations
during E3. The ground surface dominated HONO production by heterogeneous conversion of $NO_2$
during the clean episode, which was then transported throughout the column. With the increases in the
NO$_2$ mixing ratio and aerosol surface areas from the clean episode to the haze episode, the aerosol
surface production became an important nocturnal source of HONO and dominated the heterogeneous
production of HONO aloft from NO$_2$ during the haze episode.
**3.4.3 Nocturnal HONO production and loss at ground level**
The nocturnal HONO observed throughout the depth of the boundary layer is primarily from the
heterogeneous conversion of NO$_2$ on the ground surface during clean episodes. The HONO conversion
frequency can be estimated using the data from the nocturnal measurements at ground level (18:00–
06:00 LT). The heterogeneous formation of HONO in reaction (R2) is first order in NO$_2$, and the
HONO formation is proportional to the NO$_2$ concentration. The conversion frequency was derived
using the method proposed by Alicke et al. (2002). The emission ratio of HONO/NO$_x$ derived in
section 3.3 was used to correct the HONO concentration by Eq. (6). Because NO was not measured at
ground level after 14:00 on December 10[th], the NO$_x$ data was not available during the nocturnal vertical
measurements on December 10[th] and 11[th]. The average HONO$_{emis}$/HONO ratio of 35.9% ± 11.8% and
26% ± 11.3% were used to correct the observed HONO concentrations (i.e. $[HONO]_{corr} =$
$[HONO] - [HONO]_{emis}$) during the clean and the haze episodes after December 10[th], respectively. The
NO$_2$-to-HONO conversion frequency, $k_{HONO}$, can be calculated using Eq. (7), by assuming that
observed HONO comes from the conversion of NO$_2$ (Su et al., 2008a).
$$[HONO]_{corr} = [HONO] - [NO_x] \times 0.0078 \tag{6}$$
$$k_{HONO} = \frac{[HONO_{corr}]_{t_2} - [HONO_{corr}]_{t_1}}{(t_2 - t_1)[\overline{NO_2}]} \tag{7}$$
where $[\overline{NO_2}]$ is the average NO$_2$ mixing ratio during the time interval of $t_2 - t_1$. The conversion
frequencies, $k_{HONO}$, on December 9[th], 10[th], and 11[th] were 0.0082, 0.0060 and 0.0114 h$^{-1}$, respectively,
corresponding to a HONO production rate by NO$_2$ ($P_{NO_2}$) of 0.25 ± 0.03, 0.28 ± 0.02, and 0.60 ± 0.02
ppb h$^{-1}$ (i.e. $C_{HONO} \times \overline{[NO_2]}$), respectively. It is necessary to elaborate that the derived $P_{NO_2}$ is the net
HONO production, which means sources and sinks of HONO (aerosol and ground surface sources,
deposition, etc.) have already been taken into account in the $P_{NO_2}$. The HONO conversion frequency
obtained in this study is comparable to the observations by Hou et al. (2016) (clean episode: 0.0065 h$^{-1}$,
haze episode: 0.0039 h$^{-1}$) and Zhang et al. (2018) in the Beijing region (haze episode: 0.058 h$^{-1}$).
However, they are lower than the observations made by Zhang et al. (2018) during the severe haze
episode in Beijing (0.0146 h[-1]), Li et al. (2012) (0.024 ± 0.015 h[-1]) and Su et al. (2008b) (0.016 ± 0.014
h[-1]) at a rural site in southern China.
It was assumed that production of HONO on aerosol surface was insignificant compared to the
ground surface during the clean episode, which has been suggested in other studies of HONO vertical
gradient (VandenBoer et al., 2013; Wong et al., 2011; Zhang et al., 2009). Therefore, the HONO
production ($P_{NO_2}$) could be considered as a net contribution of the surface production of HONO to the
total column of HONO when HONO deposition is considered in $P_{NO_2}$. The surface production rate of
HONO of 0.25 ± 0.03 and 0.28 ± 0.02 ppb h[-1] were an order of magnitude higher than the maximum
production rate of HONO on aerosol surface (0.047 and 0.062 ppb h[-1]) on December 9[th] and 10[th]. This
result suggests that ground surface dominated HONO production by heterogeneous conversion of $NO_2$
during the clean episode. In contrast, the production of HONO solely on aerosol surface can explain the
HONO observations in the residual layer during E3, indicating that the aerosol surface production was
an active HONO production mechanism during haze episodes. The derived $P_{NO_2}$ is the total HONO
production rate of the aerosol and ground surfaces by heterogeneous conversion of $NO_2$. To compare
the HONO heterogeneous production on aerosol and ground surfaces, a deposition velocity of $NO_2$ to
the surface in the dark, $V_{dep,NO_2,}$ of 0.07 cm s[-1] (VandenBoer et al., 2013), in a boundary layer of
height, $h$ of 140 m, was used to estimate the HONO production rate by $NO_2$ on the ground surface. The
nocturnal production of HONO by heterogeneous uptake of $NO_2$ on ground surface can be estimated by
the following,
$$P_{HONO,ground} = \frac{1}{2} \frac{V_{dep,NO_2}}{h} \overline{[NO_2]}$$
(8)

The surface production rate of HONO ($P_{HONO,ground}$) was 0.47 ± 0.02 ppb h[-1] on December 11[th] (E3),
which was comparable to the HONO production rate on aerosol surface of 0.02–0.2 (0.11 ± 0.09) ppb
h[-1]. This result also suggests that the production of HONO on aerosols is an important nocturnal source
of HONO during haze episodes. The higher production rates of HONO on the ground surfaces were
consistent with the fact that the ground had a much greater surface area than the aerosol (i.e, the ground
surface area was 7140 µm[2] cm[-3] in a 140 m deep NBL, versus the average $S_{aw}$ of 2255 µm[2] cm[-3] during
E3). However, the vertical transport of the surface production of HONO throughout the column was
likely inhibited during E3. The column average concentration of HONO was independent of the mixing
ratio of HONO observed between ground level and 10 m (Fig. 10a), which may have been due to a
more stable nocturnal boundary layer structure during the haze episode.

A budget equation of nighttime HONO (Eq. 9) was utilized to separate the contributions of the

individual chemical processes involved in the nocturnal production and loss of HONO (Su et al., 2008b;
Oswald et al., 2015).
$$\frac{d[HONO]}{dt} = P_{emis} + P_{aerosol} + P_{ground} + P_A - L_{dep} \pm T_h \pm T_v \qquad (9)$$
The production terms of the HONO consist of the direct emission rate ($P_{emis}$); the heterogeneous
production rate on aerosol ($P_{aerosol}$) and ground surfaces ($P_{ground}$); and the additional nighttime
HONO source/sink ($P_A$). The loss process ($L_{dep}$) is the dry deposition rate at nighttime. $T_h$ and $T_v$
describe the horizontal and vertical transport processes, respectively. The horizontal transport, $T_h$, is
negligible in a relative calm atmosphere with low wind speeds (<1.6 m s$^{-1}$) during vertical
measurements. The vertical transport, $T_v$, acts as a sink close to the surface and as an additional source
at elevated levels. However, it is difficult to quantify $T_v$ without direct measurements of fluxes or
using the chemical transport model, and its contribution is uncertain. Without explicitly considering $T_v$,
the budget analysis is reasonable for relatively well-mixed conditions. Thus, the budget analysis is used
for the measurements conducted on December 9[th] and 10[th], when no shallow inversion layer was
observed near the surface.

Simplifying Eq. (9), the $dHONO/dt$ was approximated by $\Delta HONO/\Delta t$, which is the difference

in the observed HONO mixing ratios at two time points. An additional nocturnal production rate term
($P_A$) can be derived by Eq. (10). The emission ratio of HONO/NO$_x$ (0.78%) and HONO$_{emis}$/HONO ratio
(26% ± 11.3%) obtained in section 3.3 were used to estimate $P_{emis}$. The nocturnal production of
HONO via NO$_2$ on aerosol and ground surfaces, and the production rate terms of $P_{aerosol}$ and
$P_{ground}$ in Eq. (4) and (8) were used as representations of the nocturnal production of HONO in Eq.
(10). Here, with an overall consideration of uptake coefficient of $\gamma_{NO_2}$ used in the literature, the $\gamma_{NO_2}$
was assumed to be 5×10$^{-6}$ to estimate the HONO production rate on aerosol surface. For $L_{dep}$, the
temperature-dependent deposition velocity of HONO ($V(HONO)_T = \exp(23920/T - 91.5)$) was
used to estimate the $V_{dep,HONO}$, which decreased exponentially to non-significant values at 40 °C
(Laufs et al., 2017). The average $V_{dep,HONO}$ calculated from the nocturnal measurements (00:00–06:00
LT) was 1.8 cm s$^{-1}$, with a range of values spanning 0.9 to 3 cm s$^{-1}$, which was within the range of
previously reported values between 0.077 and 3 cm s$^{-1}$ (Harrison and Kitto, 1994; Harrison et al., 1996;
Spindler et al., 1998; Stutz et al., 2002; Coe and Gallagher, 1992; Laufs et al., 2017).
$$\frac{\Delta HONO}{\Delta t} = \frac{1}{2}\frac{V_{dep,NO_2}}{h}[NO_2] + \frac{1}{8}S_{aw}C_{NO_2}\gamma_{NO_2}[NO_2] + \frac{\Delta HONO_{emis}}{\Delta t} + P_A - \frac{V_{dep,HONO}}{h}[HONO] \quad (10)$$
Figure 11 shows the nocturnal HONO budgets from 18:00 to 06:00 LT on the 9th (C2) and 11th (E3)
of December. The production rate of HONO on aerosol surface ($0.02 \pm 0.01$ ppb h$^{-1}$) was insignificant
compared to the ground surface ($0.28 \pm 0.03$ ppb h$^{-1}$) during C2. However, an average $P_{aerosol}$ of
$0.10 \pm 0.01$ ppb h$^{-1}$ derived during E3 was comparable to the surface production rate of HONO
($P_{ground}$, $0.47 \pm 0.03$ ppb h$^{-1}$), contributing about 20% of the production of HONO, supporting the
preceding result that HONO production on aerosols was an important nocturnal source of HONO
during the haze episode. For the source of direct HONO emissions, $P_{emis}$ just provided a part of the
HONO at a rate of $0.06 \pm 0.07$ and $0.10 \pm 0.10$ ppb h$^{-1}$. The loss of HONO due to surface deposition
was the dominant sink for HONO during nighttime. The $L_{dep}$ contributed $0.74 \pm 0.31$ and $1.55 \pm 0.32$
ppb h$^{-1}$ to the nocturnal loss of HONO during C2 and E3, respectively, implying that significant
amounts of HONO were deposited to the ground surface at night. This had been suggested in another
study on the vertical gradient of HONO (VandenBoer et al., 2013).
**4 Conclusions**
High-resolution vertical profiles of HONO and NO$_2$ were measured using an IBBCEAS
instrument during the APHH-Beijing winter campaign. To the best of our knowledge, this is the first
high-resolution vertical measurements of HONO and NO$_2$ in urban areas of China. The HONO
concentrations observed during E1, C2, and E3 were $4.26 \pm 2.08$, $0.90 \pm 0.65$, and $3.54 \pm 0.91$ ppb,
respectively. A relatively well-mixed boundary layer was observed after sunset, and the vertical
distribution of HONO was consistent with reduced mixing and stratification in the lower several
hundred meters of the nocturnal urban atmosphere. The small-scale stratification of the nocturnal
atmosphere and the formation of a shallow inversion layer affected the vertical distribution of HONO
and NO$_2$. A near-steady state in HONO mixing ratio and HONO/NO$_2$ ratio was observed near midnight
on December 11th to 12th, and an approximate balance was established between the production and loss
of HONO.
Direct HONO emissions contributed an average of $29.3\% \pm 12.4\%$ to the ambient HONO levels at
night. High-resolution vertical profiles of HONO revealed (1) the ground surface dominated HONO
production by heterogeneous conversion of NO$_2$ during clean episodes, (2) the production of HONO on

aerosol surface explained the HONO observations in the residual layer during E3, suggesting that the aerosol production was an important nighttime HONO source during haze episodes. The column average HONO concentration was irrelevant to the HONO observed between the ground level and 10 m during E3, implying that the aerosols dominates the heterogeneous production of HONO aloft from $NO_2$ during haze episodes, while the surface production of HONO and direct emissions into the overlying air are minor contributors. Average dry deposition rates of $0.74 \pm 0.31$ and $1.55 \pm 0.32$ ppb h$^{-1}$ were identified during the clean and haze episodes, respectively, implying that significant amounts of HONO were deposited to the ground surface at night. Overall, these results draw a picture of the nocturnal sources of HONO during different pollution levels, and demonstrated the urgent need for high-resolution vertical measurements of HONO to a high height (e.g., using tethered balloons) and more comprehensive vertical observations to improve our understanding of the vertical distribution and chemistry of HONO in the PBL.

*Data availability*. The data used in this study are available from the corresponding author upon request (mqin@aiofm.ac.cn).

*Supplement.*

*Author contributions*. MQ and PX organized the field contributions from the Anhui Institute of Optics and Fine Mechanics group for the APHH-Beijing project. MQ and JD designed the study. WF and JD built the IBBCEAS instrument. JD and KT collected the HONO and $NO_2$ data. YS and CX provided the ancillary data. FM and MQ analyzed the data. FM wrote the paper and MQ revised it. The ranking of FM and MQ are in no particular order.

*Competing interests*. The authors declare that they have no conflict of interest.

*Special issue statement*. This article is part of the special issue "In-depth study of air pollution sources and processes within Beijing and its surrounding region (APHH-Beijing) (ACP/AMT inter-journal SI)". It is not associated with a conference.

*Acknowledgements*. We gratefully acknowledge Bin Ouyang from Cambridge University for providing HONO

measurement data for inter-comparison.

*Financial support.* This work was supported by the National Natural Science Foundation of China (41875154,
41571130023, 91544104), the National Key R&D Program of China (2017YFC0209400, 2016YFC0208204), and
the Science and Technology Major Special Project of Anhui Province, China (16030801120).

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

**Table**

**Table 1.** Classification of the meteorological conditions and corresponding concentrations of NR-PM$_1$, NO$_2$ and HONO from December 7$^{th}$ to 12$^{th}$.

| Time period | Weather condition | NR-PM$_1$ ($\mu g \cdot m^{-3}$) | HONO (ppb) | NO$_2$ (ppb) | WS (m·s$^{-1}$) | WD | $T$ (°C) | RH (%) |
|---|---|---|---|---|---|---|---|---|
| 7 Dec–8 Dec (10:00) | Haze (E1) | 30–184 | 1.49–7.59 | 24.91–65.48 | 0.03–1.95 | NW-ESE[a] | 1.6–9.3 | 36–82 |
| 8 Dec (10:00)–11 Dec | Clean (C2) | 3–97 | 0.27–3.75 | 3.33–47.84 | 0.01–6.24 | NE-NW | -2.4–9.1 | 16–53 |
| 11 Dec–12 Dec | Haze (E3) | 69–217 | 1.54–5.51 | 38.58–66.57 | 0.02–1.81 | NE-NW | -1.6–6.9 | 40–69 |

[a] NE: Northeast; ESE: East-southeast; NW: Northwest;

**Table 2.** The nocturnal gradient of HONO and NO$_2$ throughout the vertical measurements. The linear least squares regression slope and correlation coefficient of HONO and NO$_2$ to altitude identified in each vertical profile measurement.

| Date | Time period (hh:mm, LT) | Gradient-HONO (ppt m$^{-1}$) | R$^2$ | Gradient-NO$_2$ (ppt m$^{-1}$) | R$^2$ |
|---|---|---|---|---|---|
| 9/12/2016 | 22:42–23:06 | -4.56 ± 0.34 | 0.89 | -16.41 ± 1.22 | 0.89 |
| 9/12/2016 | 23:15–23:40 | -4.70 ± 0.73 | 0.65 | -18.69 ± 1.50 | 0.87 |
| 10/12/2016 | 22:36–23:01 | -0.45 ± 0.34 | 0.04 | -2.22 ± 1.23 | 0.10 |
| 10/12/2016 | 23:01–23:25 | -3.36 ± 0.52 | 0.65 | -7.59 ± 1.24 | 0.62 |
| 11/12/2016 | 22:35–23:00 | -6.92 ± 0.36 | 0.94 | -10.52 ± 0.91 | 0.86 |
| 11/12/2016 | 23:04–23:29 | -0.16 ± 0.46 | 0.006 | -5.45 ± 0.87 | 0.63 |
| 12/12/2016 | 00:00–00:26 | 0.24 ± 0.39 | 0.02 | -6.01 ± 0.69 | 0.77 |
| 12/12/2016 | 00:45–01:09 | -1.98 ± 0.28 | 0.71 | -5.70 ± 0.87 | 0.65 |

**Table 3.** Emission ratios ($\triangle$HONO/$\triangle$NO$_x$) of the fresh direct emission plumes.

| Date | Local Time | $R^2$ | $\triangle$NO/$\triangle$NO$_x$ | $\triangle$HONO/$\triangle$NO$_x$ (%) |
|---|---|---|---|---|
| 15/11/2016 | 18:05–18:15 | 0.97 | 0.99 | 1.07 |
| 16/11/2016 | 20:50–21:10 | 0.83 | 0.96 | 0.92 |
| 24/11/2016 | 20:50–21:10 | 0.92 | 1.13 | 1.12 |
| 26/11/2019 | 02:10–02:40 | 0.94 | 0.94 | 1.31 |
| 26/11/2016 | 22:15–22:30 | 0.95 | 1.00 | 1.73 |
| 28/11/2019 | 04:40–04:55 | 0.87 | 0.85 | 0.78 |
| 29/11/2016 | 03:30–03:50 | 0.95 | 0.98 | 1.60 |
| 2/12/2016 | 23:40–23:55 | 0.95 | 1.01 | 1.67 |
| 7/12/2016 | 02:25–02:35 | 0.87 | 0.90 | 1.67 |
| 10/12/2016 | 01:00–01:25 | 0.84 | 0.95 | 1.43 |
| 10/12/2016 | 02:40–02:55 | 0.86 | 0.93 | 0.79 |

**Figures**

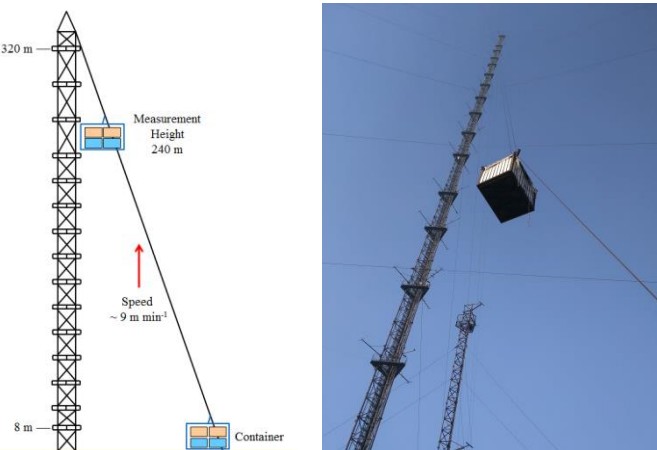

**Figure 1.** The Beijing 325-m meteorological tower (BMT) at the Institute of Atmospheric Physics (IAP).

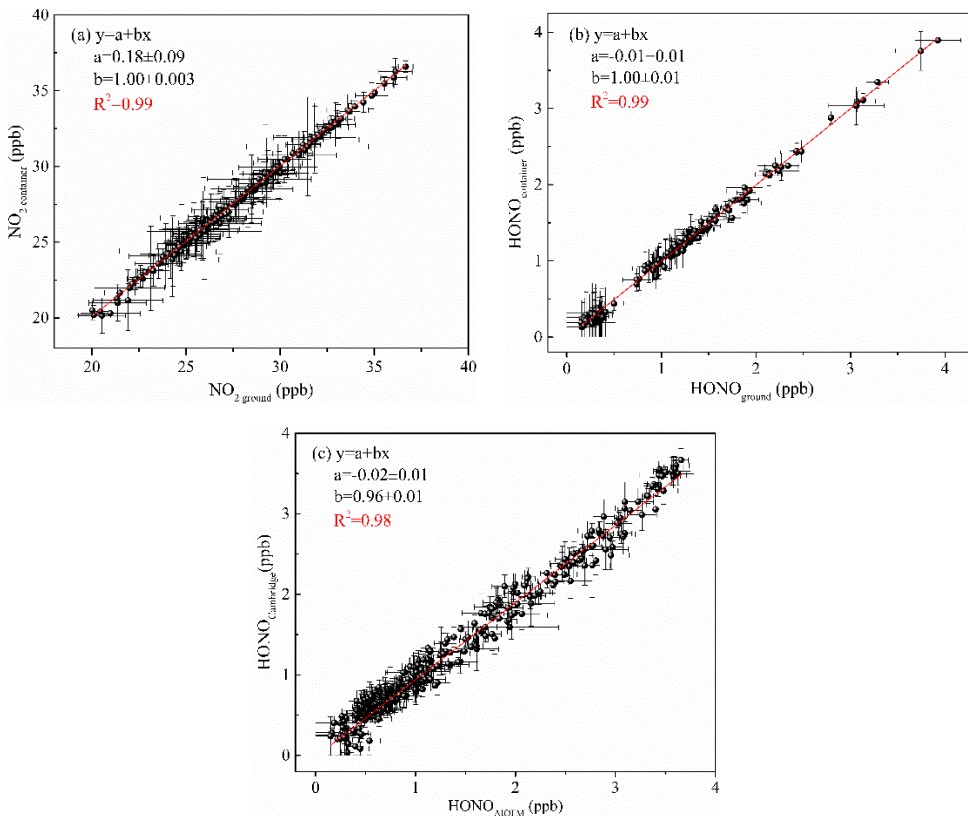

**Figure 2.** (a) Correlation of $NO_2$ concentration was measured using the two IBBCEAS instruments; (b) correlation of HONO concentration was measured using the two IBBCEAS instruments; (c) an inter-comparison between the IBBCEAS of Cambridge University and the IBBCEAS of the Anhui Institute of Optics and Fine Mechanics (AIOFM). The solid lines (red lines) show the orthogonal linear least squares regression between the two IBBCEAS instruments.

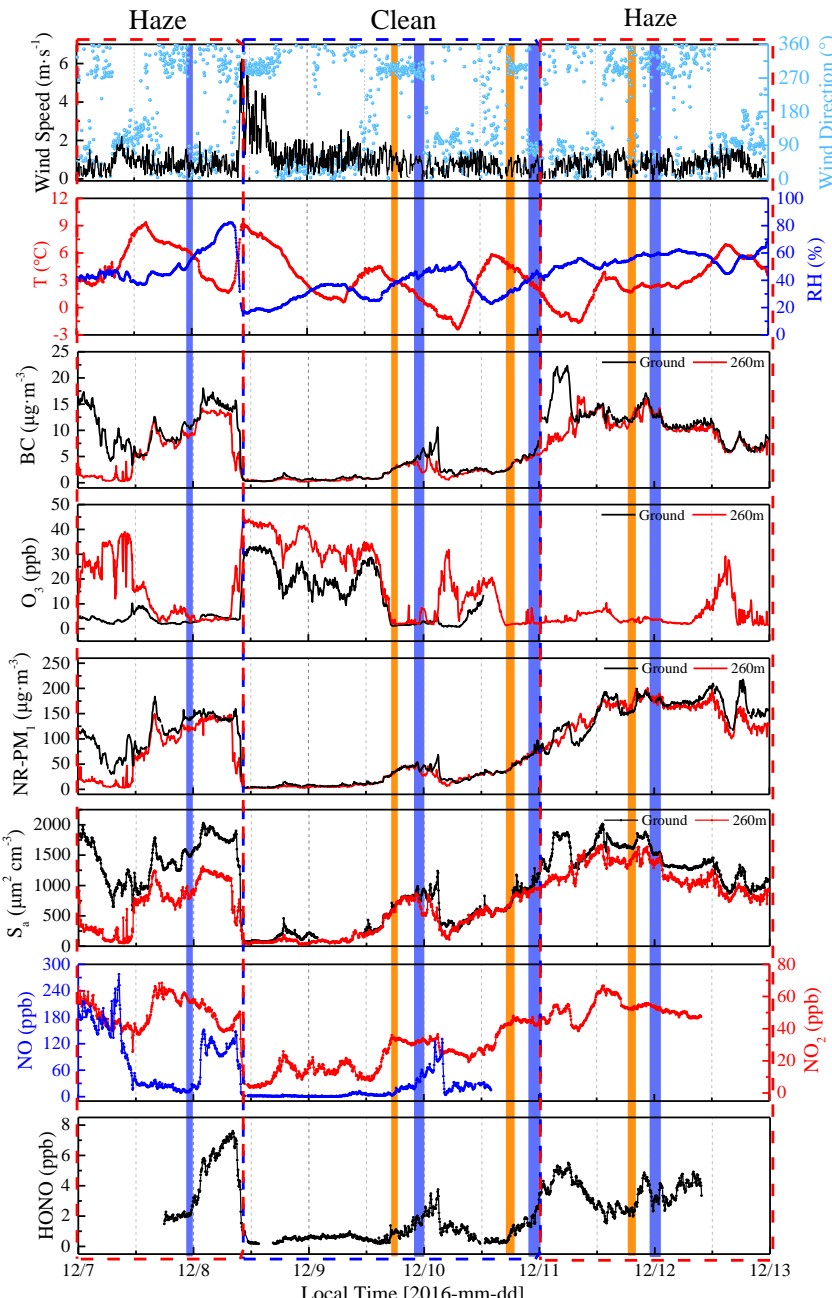

**Figure 3.** Time-series of wind speed (WS) and direction (WD), temperature (*T*), relative humidity (RH), BC, $O_3$,

NR-PM$_1$, aerosol surface area ($S_a$), NO, NO$_2$, and HONO from December 7th to 12th 2016 at the IAP-Tower

Division in Beijing, China. The measurements of NO, NO$_2$ and HONO are made at the ground level. The shaded

region represents the eight vertical measurements (Table S1). The orange shaded region represents the vertical

measurements after sunset, and the violet shaded region represents the vertical measurements at night and

midnight.

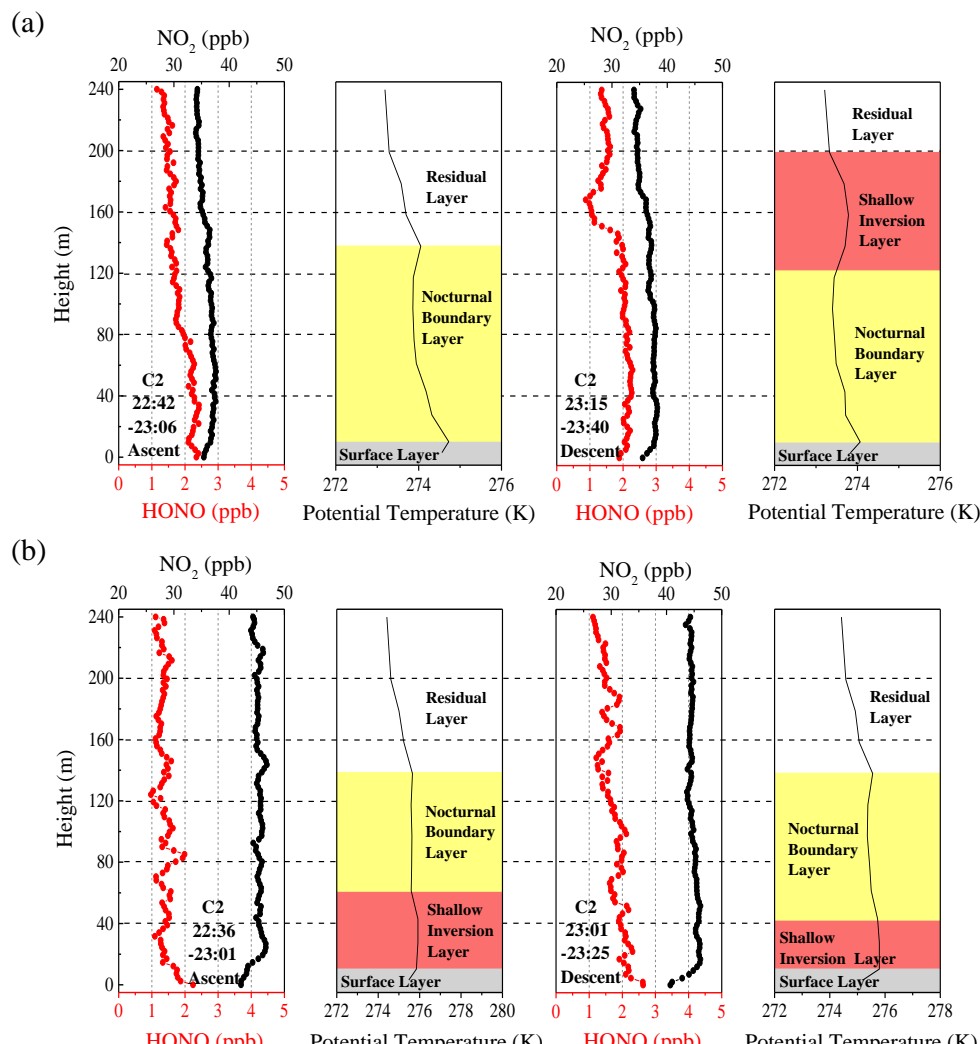

**Figure 4.** Nocturnal vertical profiles of HONO, NO₂, and the potential temperatures during the ascent and descent of the container on the (a) 9[th] and (b) 10[th] of December. The time in the figure corresponds to the measurement time of the vertical profile of the HONO and NO₂. The different colored shaded region indicates the nocturnal small-scale stratification (surface layer, nocturnal boundary layer, shallow inversion layer, and residual layer). The heights of the surface layer, the shallow inversion layer, the nocturnal boundary layer, and the residual layer are denoted by grey shaded regions, pink shaded regions, yellow shaded regions, and white shaded regions, respectively.

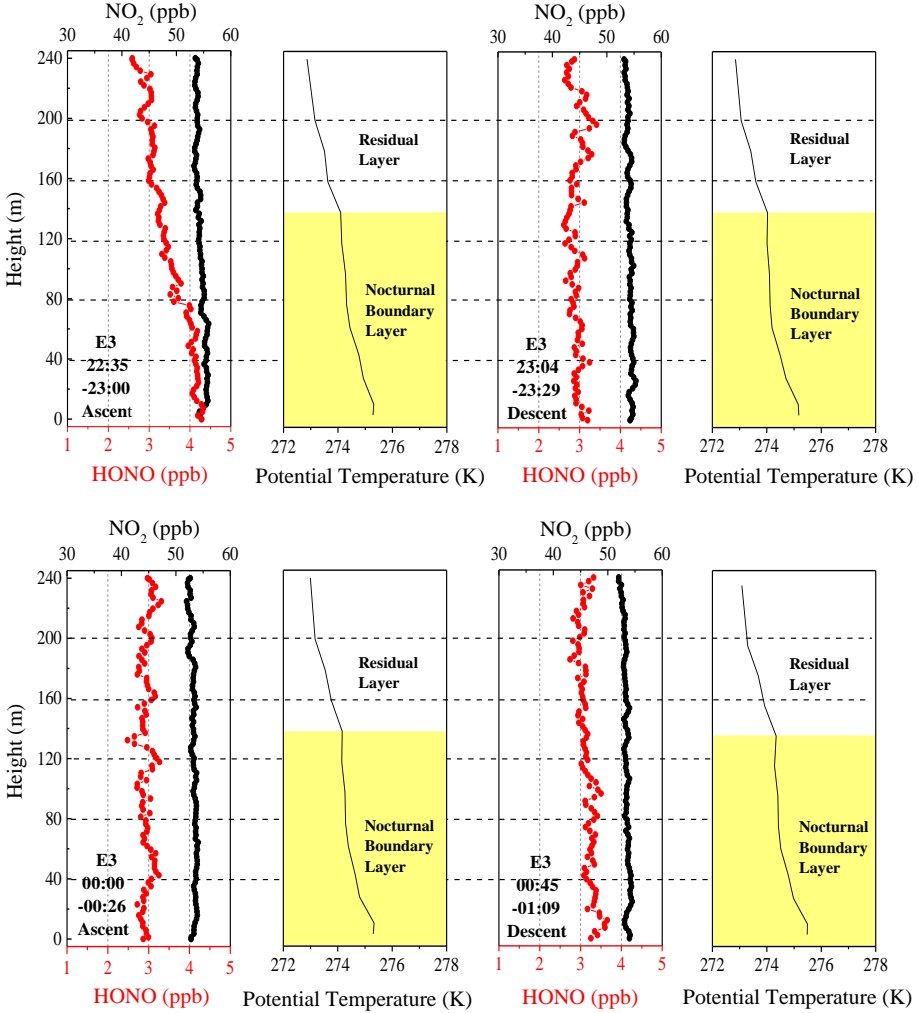

**Figure 5.** Vertical profiles of HONO and NO₂ on the night of December 11[th] and midnight of December 12[th]. The

potential temperature profiles indicate nocturnal small-scale stratification (a nocturnal boundary layer and a

residual layer). The height of the nocturnal boundary layer (NBL) is denoted by the yellow shaded region. The

time in the figure corresponds to the measurement time of the vertical profiles of HONO and NO₂.

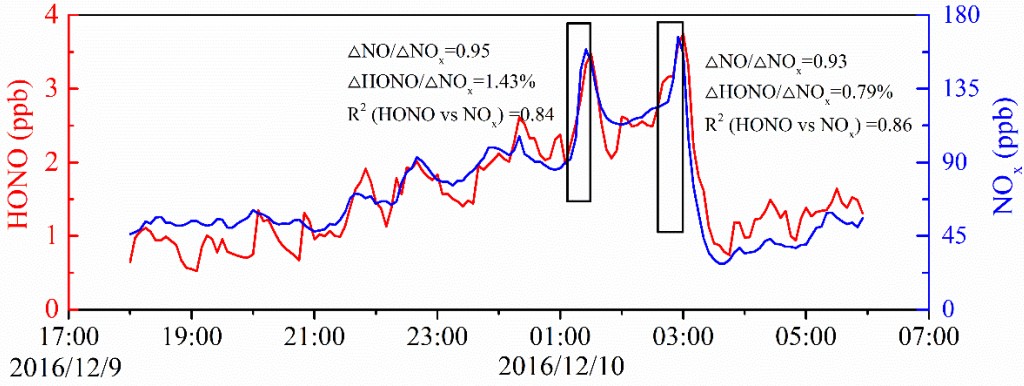

**Figure 6.** Temporal variation of nocturnal HONO and $NO_x$ on December 9$^{th}$ to 10$^{th}$, 2016. The HONO emission ratios were estimated using data collected in the black frame.

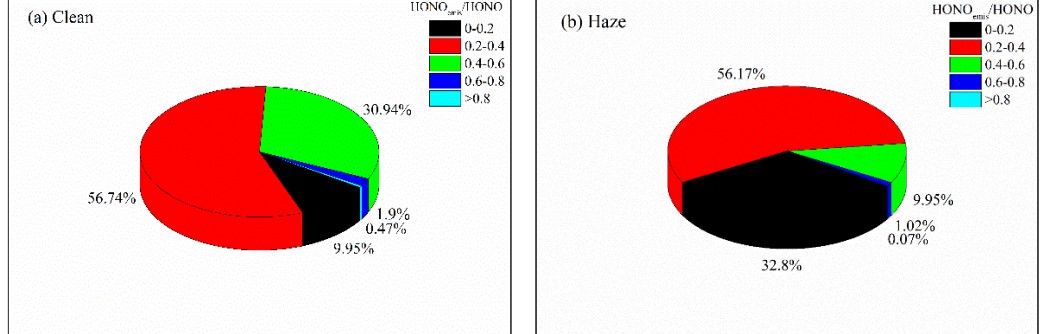

**Figure 7.** The nocturnal $HONO_{emis}$/HONO ratios frequency distribution during (a) clean and (b) haze episodes.

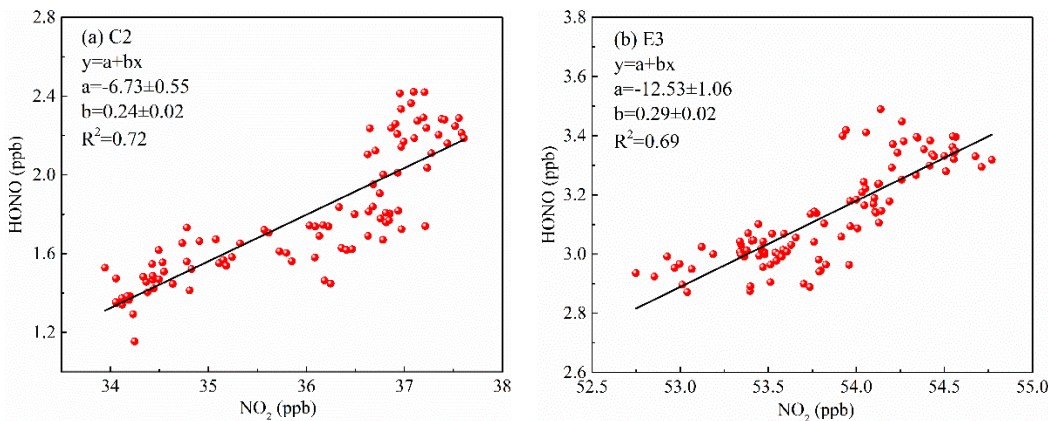

**Figure 8.** The correlation of the vertical profiles between HONO and $NO_2$ during (a) the clean episode (C2) and

(b) the haze episode (E3) using a linear least squares regression fit.

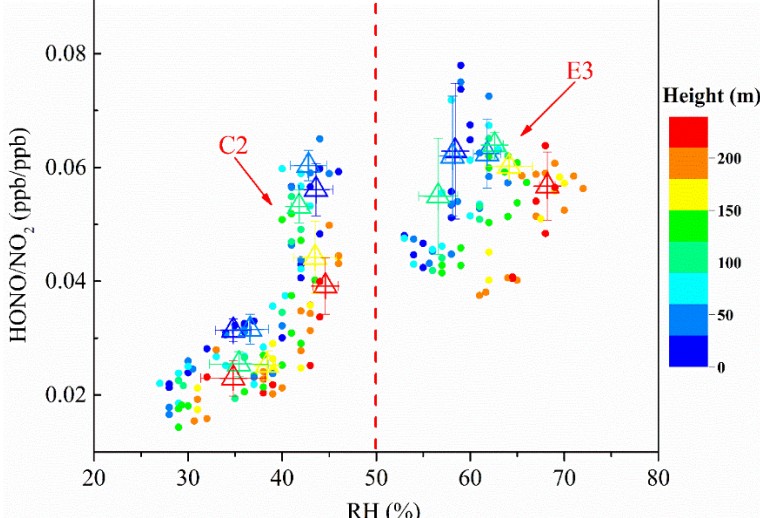

**Figure 9.** Scatter plot of HONO/$NO_2$ against RH of all vertical profiles during the clean episode (C2) and the

haze episode (E3). The HONO/$NO_2$ ratio is color coded by the heights. Triangles are the average of the first five

HONO/$NO_2$ values in each 10% RH interval at different height intervals (8–65 m, 65–120 m, 120–180 m, and

180–240 m).

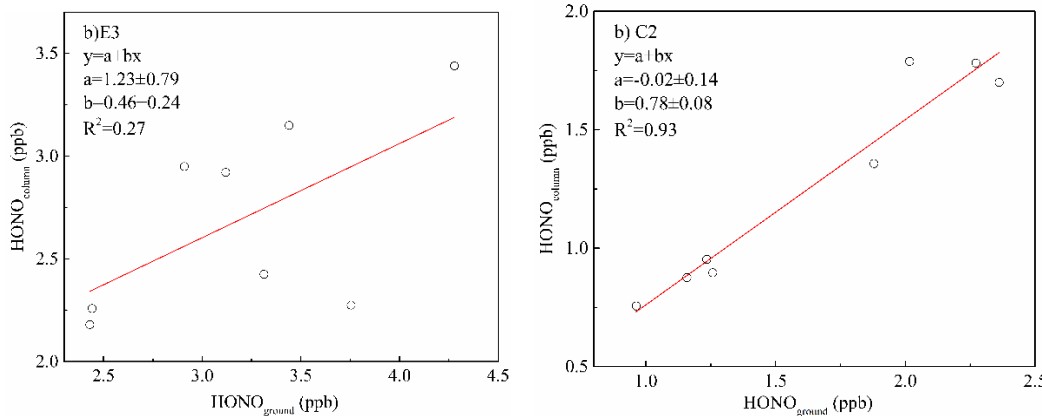

**Figure 10.** Orthogonal linear least squares correlation between the column average concentration of HONO (the average HONO column concentration from 10 to 240 m) and HONO measured from the ground level to 10 m above the ground level (AGL). Column values were calculated for (a) E3 and (b) C2.

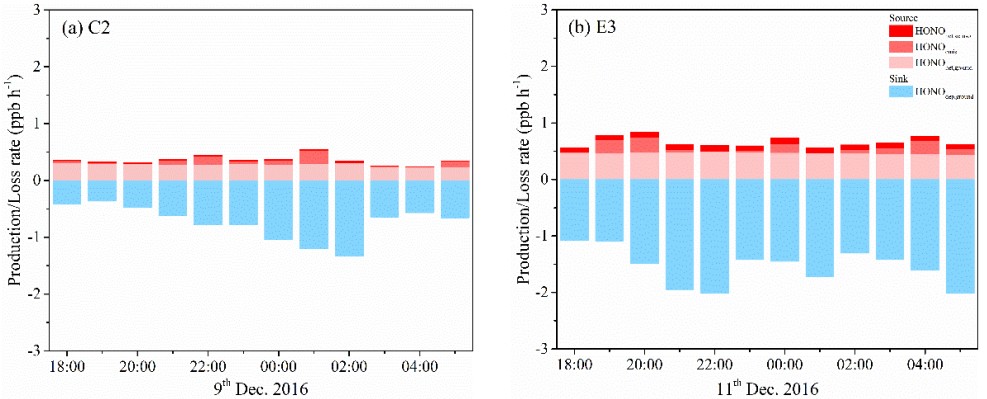

**Figure 11.** Separated contributions of production and loss terms (colored bars) of HONO on (a) the 9th (C2) and (b) 11th (E3) of December 2016.