# Peer review of "High resolution vertical distribution and sources of HONO and NO2 in the nocturnal boundary layer in urban Beijing, China"

_Atmospheric Chemistry and Physics, 2019_

## Short Comment (SC1) · 11 Sep 2019

The gradient study presented in Meng et al. is a nice piece of work showing night-time gradient data on a meteorological tower up to 250 m altitude similar to our former study (Kleffmann et al., 2003). Here a measurement container is lifted on the side wiring of the tower, far away from the open steel construction of the tower, thus minimizing potential wall effects, which is a nice idea. I strongly encourage the authors to extent these measurements including other important species for the future (see below). I have a few points which may be considered for the revised manuscript:

1) Direct HONO emissions (section 3.3):

[Figure]

The authors determined the HONO/NOx emission ratio from the minimum HONO/NOx ratio, which by definition will only result in an upper limit value, since always some secondary HONO is included in ambient measurement data. If emission ratios should be derived from field data, the authors should look for sharp peaks during night-time and should evaluate only the elevations of HONO and NOx over the background levels (DHONO/DNOx). In this case the risk overestimating direct emissions is minimized – but still even this peak data presents an upper limit caused by potential secondary HONO formation during transport from the emission to the measurement site (will be short for sharp concentration peaks). I expect that the HONO/NOx emission ratio derived by the peak method will be considerably lower, closer to direct emission data (see test stands and tunnel data: <1%). The use of correct HONO emission data is important for the whole study, since emission corrected HONO data (considerable ca. 50% contribution, typically emissions have 10-20 % contribution) is used later for the evaluation of the other HONO sources.

2) HONO formation on aerosol surfaces (section 3.4.2).

The authors used heterogeneous uptake data for NO2 derived in the laboratory to calculate potential HONO formation during night-time. However some studies cited/considered focus on the daytime production of HONO by photosensitized conversion (George et al., Sremmler et al.), leading to overestimation of the night-time conversion. Here references to other dark studies are recommended and it should be mentioned that the used value of 10ˆ-5 represents really the upper limit night-time kinetics (typically 10ˆ-6 is used in the dark). In addition, the authors should explain that they considered a 100 % yield for the NO2 conversion (see factor $\frac{1}{4}$ in eq. (1)), which is also the upper limit, e.g. when using the typically considered night-time reaction R1: 2NO2+H2O=>HONO+HNO3 (see the first reference used in line 370) for which the maximum yield is 50 %. So please define that a redox reaction (NO2+X=>HONO) is considered here. But all this will even further decrease the low contribution of particle surfaces to the night-time HONO production, which is in line with my own point of view.

[Figure]

3) Nocturnal HONO production on the ground (section 3.4.3):

Instead of using the "two-point" equation (4), simply plot the night-time HONO/NO2 ratio as a function of the time. In this case the slope determined by linear regression will statistically better describe the efficient first order NO2=>HONO conversion rate constant. Since this is a rate coefficient, it should be better termed by e.g. "k(het)" and not by the term "C" (C: concentration?), which I often found in recent Chinese HONO papers? Since HONO(corr.) will be significantly higher (see point 1) also the efficient conversion rate coefficient will increase.

In this section the authors also determine a net HONO yield from the ground surface conversion of NO2 of the order of 10 % by using deposition velocities for HONO and NO2. The value for HONO is to my opinion too low and will be more of the order of 2 cm/s (see cited studies and others). In addition please specify the value used for NO2. The HONO yield determined is quite high and our recent gradient study (Laufs et al., Atmos. Chem. Phys., 17, 6907–6923, 2017) could be also cited, where we determined lower values in the range of 0.02-0.044 by a more direct approach in good agreement with the gradient study by Stutz et al., 2002 (0.03). In addition you will find there also data for the deposition velocity of HONO, confirming a higher value, at least at the low temperatures of the present study.

4. Concept used:

The authors tried to distinguish between heterogeneous night-time formation of HONO on particle surfaces and on the ground, which is an important question with respect to the decades long discussion on the sources of HONO in the atmosphere. Here first, I am missing a more focused introduction of the basic problem. There are ground level studies, which found a nice correlation of HONO/NO2 with the particle surface area and which propose HONO formation on particles. On the other hand there are gradient studies of HONO (and NO2) which typically determine a negative gradient, pointing to the ground as most important surface. However, none of these studies can answer

the question! The correlation with particles could be also explained by the variation of the vertical mixing and the fact that both, HONO and particles are emitted/formed near the ground. Here pioneering studies by A. Febo found nice correlation of HONO with Radon during night-time and there is clearly no chemical link between both species... On the other hand the gradients could be also explained by formation on particle surfaces, if there is also a negative gradient in the particle surface area density (S/V). So what we need are gradient measurements over a few hundred meters (like the present study) but including besides HONO and its precursors (most probably $NO_2$) also the surface area density (see our former study Kleffmann et al., 2003). Only if there is no gradient in the particles, a negative gradient of the $HONO/NO_2$ ratio will show HONO formation on the ground. Thus, second, I strongly recommend that the authors use this nice tower set-up and include in future campaigns also fast particles and NO measurements!

The later is important since the titration reaction of $NO+O_3$ may mask the observation leading e.g. to artificial gradients of the $HONO/NO_2$ ratio not resulting by any HONO processes, but simply by a changing $NO_2/NO_x$ ratio. For example, the decreasing $HONO/NO_2$ ratio shown in Figure S4 (see first two gradients near the ground) is not plausible since the $HONO/NO_x$ ratio is expected to increase during the night. Most probably NO was converted into $NO_2$ leading to lower $HONO/NO_2$ ratio.

Minor points:

There are numerous errors in the references, please check, examples: Line 60: Vogel et al.; missing blanks between the references (throughout the whole manuscript). Line 68: Bröske et al., etc.

Line 76: the photocatalytic conversion of $NO_2$ on $TiO_2$ ("mineral dust") is not a redox reaction.

Line 77-82, R2: This reaction is not a photosensitized conversion.

[Figure]

Lines 84-88: In the cited studies neither all used a DOAS, nor was the DOAS installed on an elevator.

Line 93-94: In our gradient study in Chile the measurement frequency was very high, only the vertical resolution was low (should be "or").

Line 106-110: Please add our recent flux study Laufs et al., 2017 (see above).

Lines 381-384, Fig.11: I do not understand that statement. The figure shows only the column average HONO concentration is near (81%) to the ground level. That could be also explained by a particle source?

Figures 7+8: The concentrations in the residual layer typically represent daytime levels (in the absence of a volume source of HONO). Here HONO levels of 1-2 ppb are observed, which should be discussed. Such high HONO levels at 250 m altitude are really exceptional!

[Figure]

---

## Referee Comment (RC1) · Anonymous Referee #1 · 29 Oct 2019

This work by Meng et al. presents some of the first vertical profiles of HONO from China, from a tall tower located in urban Beijing. Using two IBBCEAS instruments, the Authors assess nocturnal HONO production and loss from a limited dataset of 3 case study days of HONO and NO2 observations via established approaches or other measurements for this region. Overall, this manuscript brings very little new information to bear on understanding HONO nocturnal formation in Beijing between clean and haze periods. Much of the analysis is replicated from the literature, but without using those approaches properly to place quantitative constraints on the observations. The presented dataset is frankly too limited to draw very broad conclusions, mainly because the supporting measurements presented in the methods are not being used to clearly

account for direct HONO emissions or aerosol surface area, nor for ground surface production. The interpretation of this dataset needs to be pushed further with all available measurements in order to be accepted into Atmospheric Chemistry and Physics.

One potentially productive avenue for this work would be to investigate the contribution of vertical transport of HONO produced at the surface into the nocturnal boundary layer, versus that calculated for aerosol formation. The meteorological data collected over the height of the tower is sufficient to carry out this analysis and would bring the necessary new dimension to this work. It is likely that this work can also close the mass balance of nocturnal HONO production for Beijing, but much more work is required. When combined with a thorough re-analysis of the dataset to satisfy the four major comments below, this work should be reviewed again to determine if it is suitable for acceptance.

Major Comments

1. Direct emissions: There are at least two direct emissions analyses for HONO from the Beijing vehicle fleet that have been previously published, which are cited in this work (Zhang et al 2018 and Tong et al 2016), but not used to build a deeper analysis of the observations. The Authors cite one of these and conclude that the majority of their observed nocturnal HONO comes from direct emissions. However, the Authors state they have the necessary measurements from this campaign to thoroughly quantify the HONO primary emissions (HONO, NO2, NO, and CO), but they do not use them. This analysis must be completed in full, with a figure added to this manuscript (or the SI) to show how the value stated was determined along with an assessment of its error. A thorough review and comparison to the literature for this emission ratio must also be made. It will also be valuable if the Authors can comment on the traffic patterns in Beijing that result in so much nocturnal HONO coming from direct emissions, as typically these are reported to peak during morning and evening rush hours. Finally, it is not clear if the Authors have accounted for and removed direct HONO emissions from their aerosol conversion analyses, which should be done, as it may otherwise

confound the interpretation.

2. Aerosol conversion of NO2 to HONO: A simple calculation is made to conclude that aerosol surface conversion of NO2 is not important to the observed HONO in the vertical profiles. The Authors use a single value of aerosol surface area from Beijing to perform this calculation, ignoring that they have a direct proxy for aerosol surface area from their particulate matter measurements (i.e. total PM1.0 mass). While this will also have uncertainty associated with it, the magnitude of potential error will be dramatically reduced. The current analysis is shallow and must be improved. One is left wondering why the aerosol measurements are given in the methods and not used in the data interpretation. The contribution of aerosol surface to HONO nocturnal production as a function of height has been reported infrequently and would be an improved contribution from this dataset. This should, again, include error analysis that spans the range of NO2 uptake coefficients, not only the highest value reported from the literature. The Authors are encouraged to use the chemical speciation of their aerosol to justify a selected NO2 uptake coefficient that is likely to be representative of conversion in Beijing and to propagate error based on the full range of quantified uptake coefficients towards a mass balance for HONO nocturnal production.

Further to this, the Authors make a series of confusing statements in their discussion of the importance of NO2 conversion on aerosols from their calculations. The Authors state that 1.02 ppb of HONO are produced during E3 and that is 'much less' than the measured 3.06 ppb. This is 33 %, which is certainly significant and counter to the conclusion that aerosol conversion is not important. Also here, if direct emissions are not accounted for, then this fraction becomes 1.02/1.53 which is 67 %. These findings need to be clarified as perhaps the time intervals have not been specified, which is leading to the confusion. The same analysis needs to explored and discussed for the other two vertical profiling periods using the measured vertical profile data instead of broad approximations of the necessary species.

Finally, the Authors reproduce the analysis of VandenBoer (2013) to conclude that the

data presented in Figure 11 clearly indicates the ground surface is the major HONO source, without quantifying its contribution with their measurements. Based on the presented vertical profiles in the manuscript and the SI, there is only one very clear observation of a vertical gradient that is clearly consistent with ground production (C2, Figure 7). In particular, the analysis of potential temperature to identify the nocturnal boundary layer height in Figure 8 does not show increasing HONO mixing ratios as the surface is approached, but instead there are uniform vertical profiles in the NBL and the residual layer, which indicates either efficient mixing or significant production in the overlying air (i.e. aerosol conversion). The increasing HONO mixing ratios in the residual layer matching those in the NBL over time strongly support aerosol conversion, since this air is disconnected from surface mixing if the inversion has been properly identified using the Brown et al (2012) approach. This is treated superficially in the analysis and needs to be explored in detail.

3. Nocturnal surface processes: The Authors evaluate the conversion efficiency of NO2 on the ground surface by replicating the approaches used in Zhang et al (2018) and Tong et al (2016), to find values of C-HONO that are comparable to other observations in urban environments. While this is a sound approach to interpret NO2 conversion to HONO, a quantification of the importance of the mechanism is not made. How much of the total column HONO is produced by this mechanism? The Authors have made the measurements, integrated the column HONO quantities, but not performed this analysis. Further to this, the Authors state that they calculated the amount of HONO being deposited on the surface, but do not present any of the values or a sensitivity analysis on the range of deposition velocities reported in the literature. It is concerning that a boundary layer mixing depth of 100 m was used when the mixed layer at the surface was directly measured. This section requires significant revision and expanded analysis to have sufficient quality to be accepted.

4. Figures: All measured and calculated quantities should be presented with the relevant error bars.

Figure 2 regression types are not specified in the caption or in the manuscript. The error in the measurements should be depicted on the panels and the appropriate regression for considering significant error in both measurements used (e.g. orthogonal least-distance regression). The uncertainty in the slopes and intercept should then also be reported.

The purpose of Figure 4 is very unclear. Either expand discussion around it in terms of its importance to nocturnal HONO production or remove it from the manuscript.

Figures 5 and 6 do not seem to have much purpose for this work. Figure 5 can be removed entirely as its contents are described clearly with the first and third sentences from Lines 237-240. Figure 6 could be removed entirely or one panel from parts a) and b) moved to the SI as an example since it, again, shows that the atmosphere is well mixed at sunset, which is expected and does not contribute significantly to the analysis of nocturnal sources.

The Figure 7 vertical profiles do not demonstrate that steady state between HONO production and loss has been reached, as the concentration of HONO at the surface has continued to increase. Typically, this observation has been seen after midnight (e.g. see VandenBoer et al (2013)). It does not mean that HONO deposition is not happening, but it does prevent quantifying the deposition term using the observations, which the Authors should be clear about.

Minor Comments: Line 168: If reported measurements are 15 s, then the associated detection limit should be presented.

Line 171: The measurement error was 'approximately' assessed as 9 %. How was this done and why is it 'approximately'? This should be used to denote error bars on observations presented in the figures. Have any corrections for NO2 to HONO conversion on the instrument inlet and cavity surfaces been characterized and corrected for?

Line 178: The models and detection limits of the supporting gas analyzers need to be

given. Why is none of this supporting data shown or used in the analysis?

Lines 180-185: Again, supporting measurements are noted as having been made here, but they are not used to interpret the data. These should be used as noted in the major comments above.

Lines 191-194: An $NO_2$ intercept of 750 ppt should be explained. Is it statistically different from zero? See major comment on figure regressions and error analysis above.

---

## Referee Comment (RC2) · Anonymous Referee #3 · 1 Nov 2019

General Comments. Manuscript acp-2019-613 reports results of a nighttime vertical gradient study aimed at determining the sources and sinks of nitrous acid (HONO) in Beijing, China. Measurements were made from an instrumented container capable of ascending and then descending a 325 m tower over an hour, enabling measurements at ground level and up to a height of 240 m. The maximum height achieved means that measurements covered the surface layer, nocturnal boundary layer, and residual layer. Furthermore, the measurement campaign covered three periods of time marked by clean or hazy air masses. The results are then used to draw conclusions about the relative importance of direct (automobile), ground, and aerosol sources of HONO to the airshed.

[Figure]

The analytical measurements appear to be of high quality. The tower experiments are ideal for elucidating HONO sources and sinks, and it was a good idea to operate simultaneous ground-based and vertically transported IBBCEAS systems. It is not the first time that gradient measurements of HONO and related physical/chemical have been made using a tower/elevator and much of the approach to data analysis and interpretation closely follows previous studies (especially, VandenBoer, et al. J. Geophys. Res. 2013, 118, 10,155–10,171, doi:10.1002/jgrd.50721 and Stutz et al. J. Geophys. Res. 2002, 107, doi:10.1029/2001JD000390). Despite the lack of novelty, the data has the potential to provide insights into myriad processes affecting HONO production and loss in Beijing, an area where frequent high aerosol concentrations mean that multiphase chemistry has a large influence on atmospheric composition. However, I do not feel that sufficient analysis of the data was carried out to support the authors' conclusions. Therefore, I feel that the manuscript is not ready for publication; additional work must be carried out to properly analyze the data and extract the information it holds.

Specific Comments. One of the most important aims of the manuscript is to determine the relative contribution of direct sources vs. ground and aerosol surfaces to overall HONO production at the site. The conclusions are: (1) direct sources are a major source ($\sim$51% of total ambient HONO is from combustion); (2) vertical profiles of HONO concentration do not support heterogeneous NO2-to-HONO conversion on aerosol surfaces; (3) heterogeneous NO2-to-HONO conversion on ground surfaces followed by vertical convection dominate HONO production at night. These conclusions are based on interpretation and discussion of results on p. 11-15, which I found to be unclear and not necessarily supportive of the final conclusions. To be suitable for publication I feel the relative contribution of the various HONO processes need to be quantified in more detail for the full data set and the derivations of those calculations more clearly explained in the text.

There was some ambiguity regarding how the contribution of direct emissions to ambient HONO was determined. It seems to be derived from the HONO/NOx ratio mea-

sured on days dominated by fresh vehicle emissions, which were identified when NO > 80 ppb and NO/NOx > 80%. The ratio was then used to devise a linear relationship between ambient NOx and direct HONO emissions that is applied over the whole campaign. How this relationship was used to derive the contribution of direct emissions to ambient HONO is not clear. Also, the statement that direct sources constitute 51% of total ambient HONO is generalized for the entire campaign. However, it is clear that the relative contribution of HONO sources will change with time of day, traffic intensity and type of traffic, meteorology. Indeed, it is expected that the HONO mass balance for each nocturnal profile will be different, with each source/sink having a different relative contribution depending on time and meteorology.

When determining the relative contribution of aerosol surfaces to the HONO budget, the authors attempt to quantify HONO production using an assumed uptake coefficient (for NO2-to-HONO conversion) and an aerosol surface area-to-volume ratio from a completely different study, which is supposed to be a typical of winter in Beijing. I don't see how one can justify using a single aerosol surface area-to-volume ratio for the entire campaign, especially since this will vary with time, meteorology, and day (see Table 1). Further, it is not clear why a single value was used since a suite of aerosol measurements were made during the current study; measurements of black carbon, non-refractory PM, and AMS measurements at ground and 260 m height are reported. This data should be used to refine the analysis and make it more specific for each ascent/descent period. I encourage the authors to provide a more in-depth analysis of the contribution of aerosol surface area to HONO processes. I believe there may indeed be evidence for an important role for aerosol here under hazy conditions. For example, clear gradients, denoted by decreasing HONO/NO2 ratio with elevation, were only observed on a few occasions, whereas it was more often the case that the HONO/NO2 ratio was quite constant over the vertical range. Thus, it is possible that aerosol chemistry may influence this profile, but only a detailed quantitative analysis can tell you this.

Finally, the authors state that ground level HONO production has been corrected for the influence of direct emissions from automobiles. Has this correction been carried out for the production on aerosol surfaces as well? I had a difficult time understanding the rationale of going through all the calculations in section 3.4.3 on the topic of ground level HONO production. It was not clear how equation 4 and 5 were derived. The purpose of these equations is to estimate a HONO conversion frequency, which appears to be the pseudo-first order rate constant associated with the conversion of NO2-to-HONO. These values should then be used to compare to the other HONO sources and sinks to evaluate the relative importance of all the various HONO sources/sinks. However, this comparison is not clearly carried out at the end of the paper as I would expect. Instead, starting with line 410 the text veers off subject by deriving a HONO yield from deposited NO2. While this may be a useful calculation to carry out, I don't see how it helps the authors reach their research objectives and main conclusions.

In closing, I feel the interpretation of the vertical profiles relies on highly generalized 'back of the envelope' calculations. Such calculations can be very useful in certain cases; however, the quality and quantity of the data provided by this study should allow for a more elaborate analyses. I recommend the authors revisit their data to devise a mass balance for HONO processes where they quantify the sources and sinks for each vertical profile. I expect this approach will provide valuable insights that are not clear from the current presentation of the data.

Specific comments by line number: Lines 45-133: I feel that the introduction lacks an explicit statement of the study's research objective(s). The authors provide a [somewhat long] background section reviewing aspects of HONO chemistry that ends with a summary of what they did in their study. The introduction would be far stronger if the authors include a focused discussion of what research objective they hope to achieve. This could be accomplished by stating a hypothesis, following by a plan for how they hoped to test the hypothesis. At the moment, it is only in the second to last sentence of the introduction that one finally learns that a comparison of HONO heterogeneous

chemistry on ground vs. aerosol surfaces is the major aim of the work.

64-66: The authors state that hydrolysis of NO2 on humid surfaces via R2 is considered the most likely explanation for observed HONO concentrations. The authors should be aware that this is not a viable mechanism under atmospherically relevant concentrations of NO2. Reaction R2 is too slow and only becomes important when the NO2 concentration is high enough to promote N2O4 formation. Initial mechanistic studies of this mechanism had to use high NO2 concentrations due to high instrument detection limits. More recent laboratory studies of this process using state-of-the-science sensitive field-grade instrumentation do not report this chemistry occurring under lower concentrations (50 ppb or less). One can refer to numerous lab studies over the past 10-15 years aimed at studying NO2 conversion on surfaces containing humic acid reactive sites to learn more about this. It is notable that these studies often include control experiments aimed at quantifying background levels of HONO derived from NO2 hydrolysis on their experimental system (i.e., in the absence of redox active substrates). These studies consistently show that NO2-to-HONO conversion on organic matter and other redox active surfaces produce orders of magnitude more HONO than does R2.

81: The colon after the references should be deleted.

349: The authors attribute a decrease in HONO/NO2 ratio at RH above 70% to enhanced HONO deposition. While RH does impact HONO uptake (see Donaldson et al. Environ. Sci. Technol. 2014, 48, 375), it would affect NO2-to-HONO conversion as well (by enhancing to a point); so there are competing effects here. The abovementioned Donaldson et al. paper suggests that the uptake coefficient for HONO actually decreases with increasing RH.

372: The number listed having units of h-1 is not a formation rate as stated. Those are units of a first-order rate constant. Rather, units of ppb h-1 are appropriate for a formation rate. 405: It would be beneficial to provide a visual comparison of all the production rates (calculated in units of mixing ratio per hour). See for example Fig. 11

from Vandenboer et al. JGR, 2013, 118, 10,155.

413: I recommend that the authors choose another symbol to represent the compensation point, which they call the HONO yield from deposited NO2; the greek letter phi is reserved for the photolysis quantum yield. The symbol "H" used for mixing depth is also confusing. Some readers may mistake this for the Henry's law constant without reading further into the text. Consider using "d" (for depth) or lower case "h" or "z", all of which have been used in the past to represent height.

421: Why choose a deposition velocity from the literature, which is derived from completely different studies. Isn't the point here to model the vertical profile to derive a deposition velocity and the uptake coefficients for NO2, HONO, etc. that are specific for the current sampling site?

More general remarks: More work should be devoted to providing error analysis. For example, error bars on data and errors in reported values derived from their analyses. This will be especially important when comparing quantitatively the relative contribution of sources/sinks to ambient HONO. I felt there were too many figures in the text. For example, what is the purpose of showing Figure 5, where the information is more useful plotted in the format shown in Fig. 6, etc. The manuscript could benefit from editing by a native English speaker. Grammatical errors start on the first line of the abstract and persist throughout the manuscript. Lastly, I note that the authors are comparing their IBBCEAS results to data collected on an instrument from Cambridge University. Does this contribution need to be acknowledged in some way (e.g., in the acknowledgement section or author list)?

---

## Author Comment (AC1) · 29 Jan 2020

Response to interactive comments on

**"High resolution vertical distribution and sources of HONO and NO₂ in the nocturnal boundary layer in urban Beijing, China"**

by Jörg Kleffmann.

Fanhao Meng and Min Qin

**General comments:**

*The gradient study presented in Meng et al. is a nice piece of work showing night-time gradient data on a meteorological tower up to 250 m altitude similar to our former study (Kleffmann et al., 2003). Here a measurement container is lifted on the side wiring of the tower, far away from the open steel construction of the tower, thus minimizing potential wall effects, which is a nice idea. I strongly encourage the authors to extent these measurements including other important species for the future.*

**Response:** We would like to thank Jörg Kleffmann for his time and efforts in preparing this detailed and constructive comments. We have revised the manuscript accordingly to address his concerns. Here are our point-to-point response to Jörg Kleffmann's specific comments.

**Major Comments:**

*1. Direct HONO emissions (section 3.3):*

*The authors determined the $HONO/NO_x$ emission ratio from the minimum $HONO/NO_x$ ratio, which by definition will only result in an upper limit value, since always some secondary HONO is included in ambient measurement data. If emission ratios should be derived from field data, the authors should look for sharp peaks during night-time and should evaluate only the elevations of HONO and $NO_x$ over the background levels ($DHONO/DNO_x$). In this case the risk overestimating direct emissions is minimized but still even this peak data presents an upper limit caused by potential secondary HONO formation during transport from the emission to the measurement site (will be short for sharp concentration peaks). I expect that the $HONO/NO_x$ emission ratio derived by the peak method will be considerably lower, closer to direct emission data (see test stands and tunnel data: <1%). The use of correct HONO emission data is important for the whole study, since emission corrected HONO data (considerable ca. 50% contribution, typically emissions have 10-20 % contribution) is used later for the evaluation of the other HONO sources.*

**Response:** Thank you very much for your valuable comments and suggestions. The emission factors have been derived from tunnel measurements in previous studies (Kirchstetter et al., 1996; Kurtenbach et al., 2001; Laing et al., 2017). However, considering the differences in vehicle types, fuel compositions, and other factors, the previously reported emission ratios might not be representative for Beijing region. In order to obtain the local emission factor, the HONO/NO$_x$ ratio was derived from field data. Following Jörg Kleffmann's suggestion, five criteria were applied to ensure as much of the freshly emitted air masses as possible: (a) only nighttime data (from 18:00 LT to next 6:00 LT) were included to avoid the fast photolysis of HONO; (b) only sharp peaks during nighttime and the elevations of HONO and NO$_x$ over the background levels were estimated; (c) $\Delta NO/\Delta NO_x > 0.80$; (d) good correlation between HONO and NO$_x$; (d) short duration of the plume (< 30 min). The typical nocturnal wind speed at measurement site was 1.2 m s$^{-1}$, thus the duration for fresh air masses should have been less 30 min during transport from the emission to the measurement site. Criteria (b), (c), (d) and (e) were used as indicators for identifying fresh vehicular emissions.

For a month field observation, 11 fresh plumes satisfied selection criteria (Table 1). Two examples of emission plumes selected based on the preceding criteria are shown in Fig. R1.

[Figure]

**Figure R1.** Temporal variation of nocturnal HONO and NO$_x$ on December 9$^{th}$ to 10$^{th}$, 2016. The HONO emission ratios were estimated using data collected in the black frame.

The derived HONO/NO$_x$ ratios vary from 0.78% to 1.73%, with an average value of 1.28% $\pm$ 0.36%, which was larger than the 0.53%–0.8% measured in the tunnel in Wuppertal (Kurtenbach et al., 2001), but in the range of 0.19%–2.1% reported by previous studies (Kirchstetter et al., 1996; Kurtenbach et al., 2001; Su et al., 2008; Rappengluck et al., 2013; Yang et al., 2014; Xu et al., 2015; Liang et al., 2017; Zhang et al., 2018; Li et al., 2018). Comparison of derived HONO/NO$_x$ ratios with previously reported results are summarized Table S2 in the supplementary information. To minimize the risk of

overestimating the direct HONO emissions, the minimum HONO/NO$_x$ ratio of 0.78% was used as an upper limit for the emission factor (Su et al., 2008). The direct emissions contributed an average of 29.3% ± 12.4% to the ambient HONO levels at night, with an average HONO$_{emis}$/HONO value of 35.9% ± 11.8% during the clean episode and an average HONO$_{emis}$/HONO value of 26% ± 11.3% during the haze episode. The lower vehicle emissions contribution during haze episode could be caused by an odd-even car ban, which required alternate driving days for cars with even- and odd-numbered license plates.

*2. HONO formation on aerosol surfaces (section 3.4.2).*

*The authors used heterogeneous uptake data for NO$_2$ derived in the laboratory to calculate potential HONO formation during night-time. However some studies cited/considered focus on the daytime production of HONO by photosensitized conversion (George et al., Sremmler et al.), leading to overestimation of the night-time conversion. Here references to other dark studies are recommended and it should be mentioned that the used value of 10ˆ5 represents really the upper limit night-time kinetics (typically 10ˆ-6 is used in the dark). In addition, the authors should explain that they considered a 100 % yield for the NO$_2$ conversion (see factor $\frac{1}{4}$ in eq. (1)), which is also the upper limit, e.g. when using the typically considered night-time reaction R1: 2NO$_2$+H$_2$O=>HONO+HNO$_3$ (see the first reference used in line 370) for which the maximum yield is 50 %. So please define that a redox reaction (NO$_2$+X=>HONO) is considered here. But all this will even further decrease the low contribution of particle surfaces to the night-time HONO production, which is in line with my own point of view.*

**Response:** Thanks for the comment and suggestion. The references (George et al., 2005; Stemmler et al., 2007) have been changed to the following one:

Saastad, O. W., Ellermann, T., and Nielsen, C., J.: On the adsorption of NO and NO$_2$ on cold H2O/H2SO4 surfaces, Geophys. Res. Lett., 20, 1191-1193, https://doi.org/10.1029/93GL01621, 1993.

Following the referee #1's suggestion, the NO$_2$ uptake coefficient of $1 \times 10^{-5}$ to $1 \times 10^{-6}$ in the dark were used to estimate the nighttime production of HONO from heterogeneous conversion of NO$_2$ on aerosol surface. The yield for the NO$_2$ conversion and night-time reaction have been stated in the revised manuscript as suggested. A typical heterogeneous reaction ( 2NO$_{2(g)}$+H$_2$O$_{(ads)}$

$\xrightarrow{surface}$ $HONO_{(g)}+HNO_{3(ads)}$) and the maximum yield of 50% were considered in evaluating the HONO production on aerosols, the yield of hydrolysis reaction assumes that HONO and $HNO_3$ are formed by equimolar disproportionation of two $NO_2$ molecules and immediately release HONO (Finlayson-Pitts et al., 2003; Finlayson-Pitts, 2009). The reactive uptake of $NO_2$ by the aerosol was assumed to occur on all measured aerosol surface area, regardless of chemical composition. The production of HONO on aerosol surface can be estimated using the equation of Ye et al. (2018) modified to account for the disproportionation.

The newly obtained aerosol surface areas at ground and at 260 m, which was corrected to ambient aerosol surface area ($S_{aw}$) for particle hygroscopicity via a growth factor (Liu et al., 2013; Wang et al., 2018), were used to estimate the nocturnal HONO production from heterogeneous reaction of $NO_2$ on aerosol surface. An estimate of HONO production on aerosols was made using an average observed $S_{aw}$ of 2314 $\mu m^2$ $cm^{-3}$ between 22 and 01 h (the vertical measurement periods on December 11[th]) and an upper limit of observed $NO_2$ of 54 ppb from the residual layer. The HONO production of 30-300 ppt in an interval of 1.5 h could account for HONO increases of 15-368 ppt in the residual layer between profile measurements during E3. Therefore, the production of HONO solely on aerosols can explain the HONO observations during the haze episode, implying that the aerosols plays an important role in nocturnal production of HONO during the haze episode. The same analysis was discussed for other vertical measurements during C2. The results indicated that the ground surface dominated HONO production by the heterogeneous uptake of $NO_2$ during the clean episode.

*3. Nocturnal HONO production on the ground (section 3.4.3):*

*Instead of using the "two-point" equation (4), simply plot the night-time HONO/$NO_2$ ratio as a function of the time. In this case the slope determined by linear regression will statistically better describe the efficient first order $NO_2$=>HONO conversion rate constant. Since this is a rate coefficient, it should be better termed by e.g. "k(het)" and not by the term "C" (C: concentration?), which I often found in recent Chinese HONO papers? Since HONO(corr.) will be significantly higher (see point 1) also the efficient conversion rate coefficient will increase.*

*In this section the authors also determine a net HONO yield from the ground surface conversion of*

*NO₂ of the order of 10 % by using deposition velocities for HONO and NO₂. The value for HONO is to my opinion too low and will be more of the order of 2 cm/s (see cited studies and others). In addition please specify the value used for NO₂. The HONO yield determined is quite high and our recent gradient study (Laufs et al., Atmos. Chem. Phys., 17, 6907–6923, 2017) could be also cited, where we determined lower values in the range of 0.02-0.044 by a more direct approach in good agreement with the gradient study by Stutz et al., 2002 (0.03). In addition you will find there also data for the deposition velocity of HONO, confirming a higher value, at least at the low temperatures of the present study.*

**Response:** The term "$C$" is first letter of the conversion, and we used the term "$k_{HONO}$" for conversion frequency as suggested. The conversion frequency ($k_{HONO}$) has been recalculated following the method introduced by Alicke et al. (2002). The local emission factor of 0.78% derived from field data was used to correct HONO concentration (i.e. $[HONO]_{corr} = [HONO] - [NO_x] \times 0.0078$). Since NO was not measured at ground level after 14:00 on December 10th, the average HONO$_{emis}$/HONO ratio of 35.9% $\pm$ 11.8% and 26% $\pm$ 11.3% were used to correct HONO observations during the clean and the haze episodes, respectively (i.e. $[HONO]_{corr} = [HONO] - [HONO]_{emis}$). The conversion frequencies, $k_{HONO}$, on December 9th, 10th, and 11th were 0.0082 h⁻¹, 0.0060 h⁻¹, and 0.0114 h⁻¹, respectively.

As Jörg Kleffmann suggested, the temperature-dependent deposition velocity of HONO ($V(HONO)_T = \exp(23920/T - 91.5)$) was used to estimate the dry deposition velocity of HONO (Laufs et al., 2017). The average $V_{dep,HONO}$ calculated from nocturnal observations (00:00–06:00 LT) was 1.8 cm s⁻¹, with a range of values spanning 0.9 to 3 cm s⁻¹, which was within the range of previously reported values between 0.077 and 3 cm s⁻¹ (Harrison and Kitto, 1994; Harrison et al., 1996; Spindler et al., 1998; Stutz et al., 2002; Coe and Gallagher, 1992; Laufs et al., 2017). The NO₂ deposition velocity ($V_{dep,NO_2}$) of 0.07 cm s⁻¹ (VandenBoer et al., 2013) used in calculation has been specified in the revised manuscript. Following the suggestion by the referee #1 and #3, we have significantly revised the discussion in section 3.4.3. A net HONO yield from the surface heterogeneous conversion of NO₂ has been removed in the revised manuscript. The surface production rate of HONO was calculated to compare with other HONO sources.

*4. Concept used:*

*The authors tried to distinguish between heterogeneous night-time formation of HONO on particle surfaces and on the ground, which is an important question with respect to the decades long discussion on the sources of HONO in the atmosphere. Here first, I am missing a more focused introduction of the basic problem. There are ground level studies, which found a nice correlation of $HONO/NO_2$ with the particle surface area and which propose HONO formation on particles. On the other hand there are gradient studies of HONO (and $NO_2$) which typically determine a negative gradient, pointing to the ground as most important surface. However, none of these studies can answer the question! The correlation with particles could be also explained by the variation of the vertical mixing and the fact that both, HONO and particles are emitted/formed near the ground. Here pioneering studies by A. Febo found nice correlation of HONO with Radon during night-time and there is clearly no chemical link between both species. On the other hand the gradients could be also explained by formation on particle surfaces, if there is also a negative gradient in the particle surface area density (S/V). So what we need are gradient measurements over a few hundred meters (like the present study) but including besides HONO and its precursors (most probably $NO_2$) also the surface area density (see our former study Kleffmann et al., 2003). Only if there is no gradient in the particles, a negative gradient of the $HONO/NO_2$ ratio will show HONO formation on the ground. Thus, second, I strongly recommend that the authors use this nice tower set-up and include in future campaigns also fast particles and NO measurements!*

*The later is important since the titration reaction of $NO+O_3$ may mask the observation leading e.g. to artificial gradients of the $HONO/NO_2$ ratio not resulting by any HONO processes, but simply by a changing $NO_2/NO_x$ ratio. For example, the decreasing $HONO/NO_2$ ratio shown in Figure S4 (see first two gradients near the ground) is not plausible since the $HONO/NO_x$ ratio is expected to increase during the night. Most probably NO was converted into $NO_2$ leading to lower $HONO/NO_2$ ratio.*

**Response:** We would like to thank Jörg Kleffmann for his introduction to the basic problem of heterogeneous nighttime formation of HONO and his constructive suggestions. The primary reaction surfaces for the nocturnal heterogeneous production of HONO is still controversial, and the role of the aerosols in the heterogeneous production of HONO remains an open question. In this study, we try to explain this problem by investigating the high-resolution vertical profiles of HONO at night. The aerosols as an important nocturnal source of HONO during haze episode was observed. However, this campaign only includes the limited vertical measurement data (only HONO and $NO_2$ vertical profiles).

As Jörg Kleffmann said, the vertical measurements including besides HONO and $NO_2$ also the aerosol surface areas and NO are needed to investigate the HONO chemistry, while the gradient measurement based on the nice tower set-up provides the basis for our future research. A more comprehensive vertical measurement campaign will perfomed in the future, including not only the vertical profiles of HONO and $NO_2$, but also the aerosol surface area and NO measurement.

**Minor Comments:**

*1. There are numerous errors in the references, please check, examples: Line 60: Vogel et al.; missing blanks between the references (throughout the whole manuscript). Line 68: Bröske et al., etc.*

**Response:** As suggested, we have checked and corrected the references in the revised manuscript.

*2. Line 76: the photocatalytic conversion of $NO_2$ on $TiO_2$ ("mineral dust") is not a redox reaction.*

**Response:** Following the suggestion by the referee #3, the introduction section has been revised. This statement has been removed in the revised manuscript.

*3. Line 77-82, R2: This reaction is not a photosensitized conversion.*

**Response:** Thanks for the comment. This reaction (R2) is a heterogeneous reaction, and we adjusted it to a more suitable position (line 95).

*4. Lines 84-88: In the cited studies neither all used a DOAS, nor was the DOAS installed on an elevator.*

**Response:** The statements and cited references have been revised in this paragraph, as below:

**Lines 122–128:**

"Vertical gradient observations provide evidence regarding surfaces and in situ HONO formation, which can help to understand the nighttime HONO sources. Methods of long-path differential optical absorption spectroscopy (LP-DOAS) (Stutz et al., 2002; Wong et al., 2011; Wong et al., 2012), instruments mounted on a movable elevator of a tall tower or a fixed height on a building (Kleffmann et al., 2003; VandenBoer et al., 2013; Villena et al., 2011) and aircraft measurements (Zhang et al., 2009; Li et al., 2014; Ye et al., 2018) have been applied for HONO vertical gradient observations in Europe and the America."

*5. Line 93-94: In our gradient study in Chile the measurement frequency was very high, only the vertical resolution was low (should be "or").*

**Response:** Revision has made as Jörg Kleffmann suggested.

*6. Line 106-110: Please add our recent flux study Laufs et al., 2017 (see above).*

**Response:** The introduction section of the manuscript has been revised accordingly to address the referee's comments. This paragraph (lines 163–170) have been removed in the revised manuscript, but the flux literature has been added to other parts of the manuscript.

*7. Lines 381-384, Fig.11: I do not understand that statement. The figure shows only the column average HONO concentration is near (81%) to the ground level. That could be also explained by a particle source?*

**Response:** Correlation between the column average HONO concentration and HONO measured from the ground level to 10 m in height have been reanalyzed during the clean and the haze episodes, respectively. The production of HONO on aerosols explained the HONO observations in the residual layer during E3. If the production of HONO was indeed dominated by heterogeneous reaction of $NO_2$ on aerosol and ground surfaces, the column average HONO concentration is expected to be irrelevant or relevant to the mixing ratio of HONO observed from the ground level to 10 m. Figure 10a showed that column average concentration of HONO is irrelevant to the amounts of HONO observed between the ground level and 10 m in height ($R^2 = 0.27$), which suggested that the aerosols dominated the production of HONO aloft by the heterogeneous reaction of $NO_2$ during the haze episode. However, a high correlation ($R^2 = 0.93$) between the column average HONO concentration and HONO measured from the ground level to 10 m was observed during C2 (Figure 10b), which suggested a surface HONO source at night during the clean episode. The HONO observed throughout the depth of the boundary layer was primarily derived from heterogeneous conversion of $NO_2$ on the ground followed by vertical transport throughout the column. The statements of Figure 10 (the original Figure 11) have been significantly revised in the manuscript to make our argument more clearly.

*8. Figures 7+8: The concentrations in the residual layer typically represent daytime levels (in the absence of a volume source of HONO). Here HONO levels of 1-2 ppb are observed, which should be*

*discussed. Such high HONO levels at 250 m altitude are really exceptional!*

**Response:** We sincerely thank the Jörg Kleffmann for his constructive advice. We completely agree with Jörg Kleffmann that high HONO levels in the residual layer (typically represents HONO daytime levels) are well worth studying. However, this study primarily focused on the investigation of the vertical distribution of nocturnal HONO and relative importance of aerosol and ground surfaces in nocturnal HONO production at different pollution levels. The limited vertical measurement and ancillary data restricted the analysis on daytime HONO levels. In the future, we plan to conduct further in-depth research on the daytime HONO. Combined with more comprehensive vertical measurements, the vertical distribution and sources/sinks of daytime HONO will be studied.

[revised manuscript text omitted]

---

## Author Comment (AC2) · 29 Jan 2020

Response to interactive comments on

**"High resolution vertical distribution and sources of HONO and NO₂ in the nocturnal boundary layer in urban Beijing, China"**

by Anonymous Referee #1

Fanhao Meng and Min Qin

**General comments:**

*This work by Meng et al. presents some of the first vertical profiles of HONO from China, from a tall tower located in urban Beijing. Using two IBBCEAS instruments, the Authors assess nocturnal HONO production and loss from a limited dataset of 3 case study days of HONO and NO₂ observations via established approaches or other measurements for this region. Overall, this manuscript brings very little new information to bear on understanding HONO nocturnal formation in Beijing between clean and haze periods. Much of the analysis is replicated from the literature, but without using those approaches properly to place quantitative constraints on the observations. The presented dataset is frankly too limited to draw very broad conclusions, mainly because the supporting measurements presented in the methods are not being used to clearly account for direct HONO emissions or aerosol surface area, nor for ground surface production. The interpretation of this dataset needs to be pushed further with all available measurements in order to be accepted into Atmospheric Chemistry and Physics.*

*One potentially productive avenue for this work would be to investigate the contribution of vertical transport of HONO produced at the surface into the nocturnal boundary layer, versus that calculated for aerosol formation. The meteorological data collected over the height of the tower is sufficient to carry out this analysis and would bring the necessary new dimension to this work. It is likely that this work can also close the mass balance of nocturnal HONO production for Beijing, but much more work is required. When combined with a thorough re-analysis of the dataset to satisfy the four major comments below, this work should be reviewed again to determine if it is suitable for acceptance.*

**Response:** We thank the Anonymous Referee #1 for pointing out the shortcomings in our analysis and presentation of the measurement data. We have significantly revised the manuscript to address the referee's comments and concerns. Below we give point-to-point response to address the referee's specific comments.

**Major Comments:**

*1. Direct emissions: There are at least two direct emissions analyses for HONO from the Beijing vehicle fleet that have been previously published, which are cited in this work (Zhang et al 2018 and Tong et al 2016), but not used to build a deeper analysis of the observations. The Authors cite one of these and conclude that the majority of their observed nocturnal HONO comes from direct emissions. However, the Authors state they have the necessary measurements from this campaign to thoroughly quantify the HONO primary emissions (HONO, NO₂, NO, and CO), but they do not use them. This analysis must be completed in full, with a figure added to this manuscript (or the SI) to show how the value stated was determined along with an assessment of its error. A thorough review and comparison to the literature for this emission ratio must also be made. It will also be valuable if the Authors can comment on the traffic patterns in Beijing that result in so much nocturnal HONO coming from direct emissions, as typically these are reported to peak during morning and evening rush hours. Finally, it is not clear if the Authors have accounted for and removed direct HONO emissions from their aerosol conversion analyses, which should be done, as it may otherwise.*

**Response:** As the referee suggested, we have analyzed the correlations of HONO with CO, NO, and BC, which are considered to be the primary pollutants emitted from combustion processes like vehicle emissions (Sun et al., 2014; Zhang et al., 2018). Good correlations of HONO with CO ($R^2$=0.85), NO ($R^2$=0.76), and BC ($R^2$=0.84) at ground level were observed (Figure S5). This indicated the potential impact of direct emissions on the ambient HONO concentrations at night. Following the suggestion by Jörg Kleffmann, we have re-estimated the emission factor based on the field data. Five criteria were applied to ensure as much of freshly emitted air masses as possible: (a) only nighttime data (from 18:00 LT to next 6:00 LT) were included to avoid the fast photolysis of HONO; (b) only sharp peaks during nighttime and the elevations of HONO and $NO_x$ over the background levels were estimated; (c) $\Delta NO/\Delta NO_x > 0.80$; (d) good correlation between HONO and $NO_x$; (e) short duration of the plume (< 30 min). The typical nocturnal wind speed at measurement site was 1.2 m s$^{-1}$ and the distance to the roads was less than 1 km. So the duration of fresh air masses should have been less 30 min during transport processes. Criteria (b) and (c) were used as indicators for identifying fresh vehicular emissions. Criteria (d) and (e) further confirmed that the increase in HONO was primarily caused by direct emissions instead of heterogeneous reactions of $NO_2$.

Two examples of emission plumes based on the preceding selection criteria were showed in

Figure 6 in the manuscript. The slopes of HONO to $NO_x$ can be considered as the emission ratios (Rappenglück et al., 2013). For a month field observations, 11 fresh emission plumes satisfied criteria (Table 1). The derived emission factors ranged between 0.78% and 1.73%, with an average value of 1.28% ± 0.36%, which was larger than the 0.53%–0.8% measured in the tunnel in Wuppertal (Kurtenbach et al., 2001) but within the range of previously published results (0.19%–2.1%) (Kirchstetter et al., 1996; Kurtenbach et al., 2001; Su et al., 2008; Rappengluck et al., 2013; Yang et al., 2014; Xu et al., 2015; Liang et al., 2017; Zhang et al., 2018; Li et al., 2018). The minimum ratio of 0.78% approximated the value (0.8%) measured in Wuppertal. The maximum ratio of 1.73% in our study was comparable to the value of 1.7% in Houston, Texas by Rappenglück et al. (2013). Comparisons of the derived HONO/$NO_x$ ratios with the previously reported results are summarized in Table S2 in the supplementary information, as the referee suggested.

The minimum HONO/$NO_x$ ratio of 0.78% was used to minimize the risk of overestimating direct emissions (Su et al., 2008). In this case the risk of overestimating vehicular emissions was minimized, but still could be the potential secondary HONO formation during the transport from emission to the measurement site. The direct emissions contributed an average of 29.3% ± 12.4% to the ambient HONO levels at night, with an average $HONO_{emis}$/HONO value of 35.9% ± 11.8% during the clean episode and an average $HONO_{emis}$/HONO value of 26% ± 11.3% during the haze episode. The lower vehicular emission contributions during haze episode could be caused by an odd-even car ban.

As pointed out by the referee, we have already considered the influence of direct emissions in the aerosol conversion analysis, which has been stated in detailed in section 3.4.2. The discussion in "3.3 Direct emissions" have been significantly revised to address the referee's comments and concerns.

*2. Aerosol conversion of $NO_2$ to HONO: A simple calculation is made to conclude that aerosol surface conversion of $NO_2$ is not important to the observed HONO in the vertical profiles. The Authors use a single value of aerosol surface area from Beijing to perform this calculation, ignoring that they have a direct proxy for aerosol surface area from their particulate matter measurements (i.e. total $PM_{1.0}$ mass). While this will also have uncertainty associated with it, the magnitude of potential error will be dramatically reduced. The current analysis is shallow and must be improved. One is left wondering why the aerosol measurements are given in the methods and not used in the data interpretation. The contribution of aerosol surface to HONO nocturnal production as a function of height has been*

*reported infrequently and would be an improved contribution from this dataset. This should, again, include error analysis that spans the range of NO₂ uptake coefficients, not only the highest value reported from the literature. The Authors are encouraged to use the chemical speciation of their aerosol to justify a selected NO₂ uptake coefficient that is likely to be representative of conversion in Beijing and to propagate error based on the full range of quantified uptake coefficients towards a mass balance for HONO nocturnal production.*

*Further to this, the Authors make a series of confusing statements in their discussion of the importance of NO₂ conversion on aerosols from their calculations. The Authors state that 1.02 ppb of HONO are produced during E3 and that is 'much less' than the measured 3.06 ppb. This is 33 %, which is certainly significant and counter to the conclusion that aerosol conversion is not important. Also here, if direct emissions are not accounted for, then this fraction becomes 1.02/1.53 which is 67 %. These findings need to be clarified as perhaps the time intervals have not been specified, which is leading to the confusion. The same analysis needs to explored and discussed for the other two vertical profiling periods using the measured vertical profile data instead of broad approximations of the necessary species.*

*Finally, the Authors reproduce the analysis of VandenBoer (2013) to conclude that the data presented in Figure 11 clearly indicates the ground surface is the major HONO source, without quantifying its contribution with their measurements. Based on the presented vertical profiles in the manuscript and the SI, there is only one very clear observation of a vertical gradient that is clearly consistent with ground production (C2, Figure 7). In particular, the analysis of potential temperature to identify the nocturnal boundary layer height in Figure 8 does not show increasing HONO mixing ratios as the surface is approached, but instead there are uniform vertical profiles in the NBL and the residual layer, which indicates either efficient mixing or significant production in the overlying air (i.e. aerosol conversion). The increasing HONO mixing ratios in the residual layer matching those in the NBL over time strongly support aerosol conversion, since this air is disconnected from surface mixing if the inversion has been properly identified using the Brown et al (2012) approach. This is treated superficially in the analysis and needs to be explored in detail.*

**Response:** We made a deeper analysis in aerosol conversion of NO₂ to HONO, as the referee suggested. The newly obtained aerosol surface areas at ground and at 260 m, which was corrected to ambient aerosol surface area ($S_{aw}$) for particle hygroscopicity via a growth factor (Liu et al., 2013;

Wang et al., 2018), were used to estimate the nocturnal HONO production from heterogeneous reaction of $NO_2$ on aerosol surface. The time series of $S_{aw}$ at ground level and at 260 m is shown in Figure. S2 in the supplementary information. The surface area information for particles larger than 0.5 µm were not valid at ground level and at 260 m during the measurement periods. Hence, this is a lower limit estimate of the total surface area for the heterogeneous reaction. The vertical profiles of HONO and $NO_2$, and the aerosol surface area measured at 260 m allowed us to estimate the nocturnal HONO production by heterogeneous uptake of $NO_2$ on aerosol surface.

As the referee pointed out, the influence of direct HONO emissions have been considered in the aerosol conversion analysis. The CO and BC measured at ground level were independent of the CO and BC observed at 260 m during the haze period (Fig. S10), since it can be expected that air masses in the residual layer were decoupled from the ground-level processes and largely free of $NO_2$ emissions (Brown et al., 2012; VandenBoer et al., 2013). An estimate of the nocturnal HONO production on aerosol surface was made using the RH corrected aerosol surface area ($S_{aw}$) and $NO_2$ observations from the residual layer during E3. The yield of hydrolysis reaction assumes that HONO and $HNO_3$ are formed by equimolar disproportionation of two $NO_2$ molecules and immediately release HONO (Finlayson-Pitts et al., 2003; Finlayson-Pitts, 2009). The reactive uptake of $NO_2$ by the aerosols was assumed to occur on all measured aerosol surface area, regardless of chemical composition. An $NO_2$ uptake coefficients in the dark of $1 \times 10^{-5}$ to $1 \times 10^{-6}$ from the literature, an average $S_{aw}$ of 2314 µm$^2$ cm$^{-3}$ observed in the residual layer between 22 and 01 h (the vertical measurement periods on December 11[th]), and the upper limit of the observed $NO_2$ of 54 ppb from the residual layer were used during E3. The HONO production of 30-300 ppt in an interval of 1.5 h could account for HONO increases of 15-368 ppt between profile measurements. Thus, the production of HONO solely on aerosol surface explained HONO observations during E3. In addition, the data presented in Figure 10 (the original Figure 11) were divided into the clean episode and the haze episodes for re-analysis. The column average concentration of HONO was independent of the HONO mixing ratio observed from the ground level to 10 m in height ($R^2 = 0.27$) during E3 (Figure 10a), suggesting that the aerosols presumably dominates the production of HONO aloft by heterogeneous conversion of $NO_2$ during the haze episode.

The same analysis for HONO production on aerosol surface were also made to the other vertical measurements during C2, as the referee suggested. A high correlations ($R^2 = 0.83$) between the

measured CO and BC at ground level and the CO and BC at 260 m were observed (Figure. S10), which indicated that vehicle emissions affected air masses in the residual layer. The lack of NO vertical profile cannot directly correct the contribution of direct HONO emissions. So we assumed that the contribution of direct emissions was consistent at ground level and in the residual layer, and the relative importance of aerosol and ground surfaces in nocturnal HONO production could be roughly estimated. The average $HONO_{emis}/HONO$ ratio of 35.9% ± 11.8% during the clean episode was used as the contribution of direct emissions to ambient HONO levels during C2. This is a higher limit estimate of the contribution of direct emissions to the HONO levels in the residual layer. The average $S_{aw}$ of 791 and 894 µm$^2$ cm$^{-3}$ from 17 to 24 h (the vertical measurement periods on December 9$^{th}$ and 10$^{th}$), and the upper limit of NO$_2$ observations of 36 and 44 ppb from the residual layer were used to estimate HONO production on aerosols on December 9$^{th}$ and 10$^{th}$. The HONO production of 26–259 ppt in an interval of 5.5 h was lower than the direct emissions corrected HONO increases of 305–608 ppt between the vertical profile measurements on December 9$^{th}$. The formation of HONO solely on aerosol surface cannot explain the observed HONO increases in the residual layer. The HONO observed in the residual layer was primarily derived from heterogeneous conversion of NO$_2$ on the ground surface followed by vertical transport throughout the column. The column average concentration of HONO was related to the amounts of HONO observed between the ground level and 10 m (Figure 10b, $R^2$ = 0.93), which also suggested that the surface source of HONO affected HONO observed throughout the depth of boundary layer. However, the HONO production on aerosols in an interval of 5.35 h is 33–332 ppt on December 10$^{th}$, which was comparable to the direct emissions corrected HONO increases of 114–369 ppt between the two vertical profile measurements. This could have been due to the formed shallow inversion layer near the surface (Figure 4), which inhibited the vertical transport of nighttime HONO at ground level. Besides the contribution of direct HONO emissions could be even more overestimated, which might also affect the estimated result. All of these may result in the aerosols dominated the heterogeneous production of HONO from NO$_2$ in the residual layer on December 10$^{th}$.

As the referee suggested, we have significantly modified the discussion in this section to make our statements more clearly. The representative uptake coefficient of NO$_2$ in Beijing derived from the chemical speciation of the aerosol will be further studied in other manuscript. In this study, we primarily focused on estimating the relative importance of aerosol and ground surface in nocturnal

HONO production at different pollution levels.

*3. Nocturnal surface processes: The Authors evaluate the conversion efficiency of NO$_2$ on the ground surface by replicating the approaches used in Zhang et al (2018) and Tong et al (2016), to find values of C-HONO that are comparable to other observations in urban environments. While this is a sound approach to interpret NO$_2$ conversion to HONO, a quantification of the importance of the mechanism is not made. How much of the total column HONO is produced by this mechanism? The Authors have made the measurements, integrated the column HONO quantities, but not performed this analysis. Further to this, the Authors state that they calculated the amount of HONO being deposited on the surface, but do not present any of the values or a sensitivity analysis on the range of deposition velocities reported in the literature. It is concerning that a boundary layer mixing depth of 100 m was used when the mixed layer at the surface was directly measured. This section requires significant revision and expanded analysis to have sufficient quality to be accepted.*

**Response:** Thanks for the comment and suggestion. Following Jörg Kleffmann's suggestion, the slope determined by linear regression will statistically better describe the efficient first order conversion rate constant of NO$_2$ to HONO. Thus, we re-estimated the conversion frequency and corresponded HONO production rate by NO$_2$ ($P_{NO_2}$). Direct HONO emissions have been considered and removed in evaluating the conversion frequency. The conversion frequency during vertical measurements (December 9[th], 10[th], and 11[th]) were 0.0082, 0.0060 and 0.0114 h$^{-1}$, respectively, corresponding to a HONO production rate by NO$_2$ of $0.25 \pm 0.03$, $0.28 \pm 0.02$, and $0.60 \pm 0.02$ ppb h$^{-1}$. It is necessary to elaborate that the derived $P_{NO_2}$ is net HONO production, which means sources and sinks of HONO (aerosol and ground surface source, deposition, etc.) have already been taken into account in $P_{NO_2}$. As the referee suggested, the net contribution of surface production of HONO to the column could be roughly estimated based on the assumption that HONO production on aerosols was insignificant compared to the ground surface during the clean episode, which has been suggested in other studies of HONO vertical gradient (VandenBoer et al., 2013; Wong et al., 2011; Zhang et al., 2009). The surface production rate of HONO was estimated to be $0.25 \pm 0.03$ and $0.28 \pm 0.02$ ppb h$^{-1}$ on December 9[th] and 10[th], respectively, which was an order of magnitude higher than the maximum production rate of HONO on aerosols (0.047 and 0.062 ppb h$^{-1}$). The agreement supported that ground surface dominated HONO production by heterogeneous conversion of NO$_2$ during the clean episode.

In contrast, the production of HONO on aerosols as an important nighttime HONO source cannot be ignored during the haze episode, as discussed in section 3.4.2. To estimate the contribution of the surface HONO production, a deposition velocity of $NO_2$ to the surface in the dark, $V_{dep,NO_2}$ of 0.07 cm s$^{-1}$ (VandenBoer et al., 2013), in a measured boundary layer of height, $h$ of 140 m, were used to estimate the HONO production rate ($P_{HONO,ground}$) by heterogeneous conversion of $NO_2$ on ground surfaces ($P_{HONO,ground} = \frac{1}{2}\frac{V_{dep,NO_2}}{h}\overline{[NO_2]}$). The $P_{HONO,ground}$ of 0.47 ± 0.02 ppb h$^{-1}$ on December 11$^{th}$ (E3) was comparable to the HONO production rate on aerosol surface of 0.2 ppb h$^{-1}$. This result also suggested that the production of HONO on aerosols is an important nocturnal HONO source during the haze episode.

The calculation and sensitivity analysis of deposition velocity of HONO have been stated in the revised manuscript, as the referee suggested. The temperature-dependent deposition velocity of HONO ($V(HONO)_T = \exp(23920/T - 91.5)$) (Laufs et al., 2017) was used to estimate the deposition velocity of HONO ($V_{dep,HONO}$). The average $V_{dep,HONO}$ calculated from nocturnal measurements (00:00–06:00 LT) was 1.8 cm s$^{-1}$, with a range of values spanning 0.9 to 3 cm s$^{-1}$, which was within the range of previously reported values between 0.077 and 3 cm s$^{-1}$ (Harrison and Kitto, 1994; Harrison et al., 1996; Spindler et al., 1998; Stutz et al., 2002; Coe and Gallagher, 1992; Laufs et al., 2017). A mesured boundary layer mixing height of 140 m was also used for analysis as suggested. Following the referree's suggestion, we also made a budget analysis of nighttime HONO, which was utilized to estimate nocturnal sources/sinks of HONO. The average dry deposition rate ($L_{dep}$) of HONO was estimated to be 0.74 ± 0.31 and 1.55 ± 0.32 ppb h$^{-1}$ during C2 and E3, respectively, implying that significant amounts of HONO were deposited to the ground surface at night. The budget analysis results also suggested that the production rate of HONO on aerosols (0.19 ± 0.01 ppb h$^{-1}$) was comparable to the surface production rate (0.47 ± 0.03 ppb h$^{-1}$) during the haze episode.

We have significantly revised the discussion in the manuscript in "3.3 Direct emissions", "3.4.2 Relative importance of aerosol and ground surfaces in nocturnal HONO production" and "3.4.3 Nocturnal HONO production and loss at ground level" to address referee's concerns and to make our argument more clearly.

*4. Figures: All measured and calculated quantities should be presented with the relevant error bars.*

*Figure 2 regression types are not specified in the caption or in the manuscript. The error in the measurements should be depicted on the panels and the appropriate regression for considering significant error in both measurements used (e.g. orthogonal least-distance regression). The uncertainty in the slopes and intercept should then also be reported.*

*The purpose of Figure 4 is very unclear. Either expand discussion around it in terms of its importance to nocturnal HONO production or remove it from the manuscript.*

*Figures 5 and 6 do not seem to have much purpose for this work. Figure 5 can be removed entirely as its contents are described clearly with the first and third sentences from Lines 237-240. Figure 6 could be removed entirely or one panel from parts a) and b) moved to the SI as an example since it, again, shows that the atmosphere is well mixed at sunset, which is expected and does not contribute significantly to the analysis of nocturnal sources.*

*The Figure 7 vertical profiles do not demonstrate that steady state between HONO production and loss has been reached, as the concentration of HONO at the surface has continued to increase. Typically, this observation has been seen after midnight (e.g. see VandenBoer et al (2013)). It does not mean that HONO deposition is not happening, but it does prevent quantifying the deposition term using the observations, which the Authors should be clear about.*

**Response:** Thanks for the comment and suggestion. The relevant error bars have been presented in the revised manuscript as suggested. We have significantly modified Figure 2 and added the regression type to the caption. The orthogonal linear squares regression was utilized to estimate the error of both measurements. The uncertainty of slope and intercept were also added in Figure 2. As the referee pointed out, the measurement uncertainty of IBBCEAS instrument have already been depicted in section 2.2 (line 249–253) of the manuscript. Thus, we did not add the statements of measurement uncertainty of IBBCEAS instrument here.

The purpose of Figure 4 was to identify the variation of wind speed in the overlying air. The vertical profile data were used for analysis when wind speed less than 6 $m \cdot s^{-1}$ throughout the column. The statements in line 334–338 are sufficient to describe its contents, so we removed it from the manuscript as suggested. Figure 5 has also been removed, and the panel (a) of Figure 6 has been moved to the supplement as an example, as the referee suggested.

We agree that a near-steady state between HONO production and loss has not been reached in

Figure 4 (the original Figure 7). A near-steady state plateau in HONO mixing ratio and HONO/NO$_2$ was observed near midnight during E3 in Figure 5 (the original Figure 8), and the vertical profiles showed an approximate plateau between 23 to 01 h. We have revised the description of Figure 4 and 5 (the original Figure 7 and 8) in section 3.2.2 to make our argument more clearly.

**Minor Comments:**

*1. Line 168: If reported measurements are 15 s, then the associated detection limit should be presented.*

**Response:** The detection limits of HONO and NO$_2$ at time resolution of 15s are 120 ppt and 200 ppt, respectively, which have been added to the revised manuscript (line 247–248) as suggested.

*2. Line 171: The measurement error was 'approximately' assessed as 9 %. How was this done and why is it 'approximately'? This should be used to denote error bars on observations presented in the figures. Have any corrections for NO$_2$ to HONO conversion on the instrument inlet and cavity surfaces been characterized and corrected for?*

**Response:** The measurement error of 9% is the total relative uncertainty of the IBBCEAS instrument, which is estimated considering the uncertainty in the absorption cross sections (5%), the calibration of mirror reflectivity (5%), spectral fitting (4%), the correction of effective cavity length (3%), the pressure in the cavity (1%), $\Delta I/I_0$ (1%) and sample loss (0.5%). The propagated uncertainties are estimated to be 8.7% which is approximately to be 9%. The relative uncertainty of IBBCEAS instrument and estimated uncertainty of 8.7% have been presented in the revised manuscript.

The inlet and optical cavity made exclusively of PFA Teflon are used to minimize the generation and wall loss of HONO. To investigate any potential secondary HONO formation on the inlet or cavity walls from conversion of NO$_2$. We measured 80 ppb NO$_2$ at different RH levels (20% RH, 30% RH, 50% RH and 70% RH) flowing through a 3 m PFA inlet tube into the IBBCEAS instrument for a long time at typical sampling flow rate (6 L min$^{-1}$). No significant HONO was observed in the cavity, suggesting that the secondary HONO formation is negligible for IBBCEAS instrument under typical operation condition. A more detailed description of our IBBCEAS instrument can be found in our published article Du et al. (2018).

*3. Line 178: The models and detection limits of the supporting gas analyzers need to be given. Why is none of this supporting data shown or used in the analysis?*

**Response:** Instrumental section has been revised including adding the models and detection limits of gas analyzers and giving related references. The auxiliary data was also shown in manuscript Figure 3, which was used in the subsequent analysis as the referee suggested.

*4. Lines 180-185: Again, supporting measurements are noted as having been made here, but they are not used to interpret the data. These should be used as noted in the major comments above.*

**Response:** The auxiliary data have been used to interpret the vertical measurements in the subsequent analysis.

*5. Lines 191-194: An $NO_2$ intercept of 750 ppt should be explained. Is it statistically different from zero? See major comment on figure regressions and error analysis above.*

**Response:** We would first like to apologize for our mistake. The $NO_2$ intercept of 750 ppt in Figure 2 was caused by the mismatch of the timeline of $NO_2$ data measured by two IBBCEAS in correlation analysis. We have corrected this error and significantly modified Figure 2 as the referee suggested.

[revised manuscript text omitted]

---

## Author Comment (AC3) · 29 Jan 2020

Response to interactive comments on

**"High resolution vertical distribution and sources of HONO and NO₂ in the nocturnal boundary layer in urban Beijing, China"**

by Anonymous Referee #3

Fanhao Meng and Min Qin

**General comments:**

*Manuscript acp-2019-613 reports results of a nighttime vertical gradient study aimed at determining the sources and sinks of nitrous acid (HONO) in Beijing, China. Measurements were made from an instrumented container capable of ascending and then descending a 325 m tower over an hour, enabling measurements at ground level and up to a height of 240 m. The maximum height achieved means that measurements covered the surface layer, nocturnal boundary layer, and residual layer. Furthermore, the measurement campaign covered three periods of time marked by clean or hazy air masses. The results are then used to draw conclusions about the relative importance of direct (automobile), ground, and aerosol sources of HONO to the airshed.*

*The analytical measurements appear to be of high quality. The tower experiments are ideal for elucidating HONO sources and sinks, and it was a good idea to operate simultaneous ground-based and vertically transported IBBCEAS systems. It is not the first time that gradient measurements of HONO and related physical/chemical have been made using a tower/elevator and much of the approach to data analysis and interpretation closely follows previous studies (especially, VandenBoer, et al. J. Geophys. Res. 2013, 118, 10,155–10,171, doi:10.1002/jgrd.50721 and Stutz et al. J. Geophys. Res. 2002, 107, doi:10.1029/2001JD000390). Despite the lack of novelty, the data has the potential to provide insights into myriad processes affecting HONO production and loss in Beijing, an area where frequent high aerosol concentrations mean that multiphase chemistry has a large influence on atmospheric composition. However, I do not feel that sufficient analysis of the data was carried out to support the authors' conclusions. Therefore, I feel that the manuscript is not ready for publication; additional work must be carried out to properly analyze the data and extract the information it holds.*

**Response:** We thank Anonymous Referee #3 for the valuable comments. The detailed and insightful comments and suggestions have greatly helped us to revise and improve the manuscript. We would like to point out that the measurement container in our study is lifted on the side wiring of the tower,

far away from the steel structure of the tower, thus minimizing potential wall effects, which is different from previous studies (Kleffmann et al., 2003; VandenBoer et al., 2013). To the best of our knowledge, this study is the first high-resolution vertical measurements conducted in China, which has suffered from heavy air pollution for several years. We have significantly revised the manuscript accordingly to address the referee's comments and concerns. Below are a point-to-point response to address the referee's specific comments.

**Specific comments:**

*One of the most important aims of the manuscript is to determine the relative contribution of direct sources vs. ground and aerosol surfaces to overall HONO production at the site. The conclusions are: (1) direct sources are a major source (~51% of total ambient HONO is from combustion); (2) vertical profiles of HONO concentration do not support heterogeneous $NO_2$-to-HONO conversion on aerosol surfaces; (3) heterogeneous $NO_2$-to-HONO conversion on ground surfaces followed by vertical convection dominate HONO production at night. These conclusions are based on interpretation and discussion of results on p. 11-15, which I found to be unclear and not necessarily supportive of the final conclusions. To be suitable for publication I feel the relative contribution of the various HONO processes need to be quantified in more detail for the full data set and the derivations of those calculations more clearly explained in the text.*

*There was some ambiguity regarding how the contribution of direct emissions to ambient HONO was determined. It seems to be derived from the $HONO/NO_x$ ratio measured on days dominated by fresh vehicle emissions, which were identified when $NO > 80$ ppb and $NO/NO_x > 80\%$. The ratio was then used to devise a linear relationship between ambient $NO_x$ and direct HONO emissions that is applied over the whole campaign. How this relationship was used to derive the contribution of direct emissions to ambient HONO is not clear. Also, the statement that direct sources constitute 51% of total ambient HONO is generalized for the entire campaign. However, it is clear that the relative contribution of HONO sources will change with time of day, traffic intensity and type of traffic, meteorology. Indeed, it is expected that the HONO mass balance for each nocturnal profile will be different, with each source/sink having a different relative contribution depending on time and meteorology.*

**Response:** Thanks for the comment and suggestion. Considering the differences in the type of

vehicles, fuel compositions, etc., the reported emission factors of HONO might not be representative for the Beijing region. The local emission factors of HONO were derived from the field data. Following the suggestions by Jörg Kleffmann, five criteria were applied to ensure as much of the freshly emitted air masses as possible: (a) only nighttime data (from 18:00 LT to next 6:00 LT) were included to avoid the fast photolysis of HONO; (b) only sharp peaks during nighttime and the elevations of HONO and $NO_x$ over the background levels were estimated; (c) $\Delta NO/\Delta NO_x > 0.80$; (d) good correlation between HONO and $NO_x$; (e) short duration of the plume ($< 30$ min). Criteria (b) and (c) were used as indicators for identifying fresh vehicular emissions. Criteria (d) and (e) further confirmed that the increase in HONO was primarily caused by freshly emitted plumes instead of heterogeneous reactions of $NO_2$.

For a month measurements, 11 fresh plumes satisfied selection criteria (Table 1). Two examples of fresh plumes observed on December 9th to 10th, 2016 based on the preceding selection criteria were shown in Figure 6 in the manuscript. The derived emission factors varied from 0.78% to 1.73%, with an average value of 1.28% ± 0.36%, which was in the range of previously published results (0.19%–2.1%) (Kirchstetter et al., 1996; Kurtenbach et al., 2001; Su et al., 2008; Rappengluck et al., 2013; Yang et al., 2014; Xu et al., 2015; Liang et al., 2017; Zhang et al., 2018; Li et al., 2018). Comparisons of derived HONO/$NO_x$ ratios with the previously reported results were summarized in Table S2 in the supplementary information. To minimize the risk of overestimating direct emissions, the minimum HONO/$NO_x$ ratio was used as an upper limit for the emission factor (Su et al., 2008). The HONO/NOx ratio of 0.78% was used to estimate the direct HONO emissions ($[HONO]_{emis} = 0.0078 \times [NO_x]$). The average HONO$_{emis}$/HONO ratio was used to evaluate the contribution of direct emissions to ambient HONO levels at night. The vehicle emissions contributed 29.3% ± 12.4% to the ambient HONO concentrations at night. As the referee pointed out, we have considered the difference in direct HONO emissions at different pollution levels, the contribution of direct emissions during the clean and haze episodes were estimated respectively. The contributions of direct emissions to ambient HONO levels were estimated to be 35.9% ± 11.8% and 26% ± 11.3% during the clean and the haze episodes, respectively. The lower vehicle emissions contribution during the haze episode could have been caused by an odd-even car ban, which required alternate driving days for cars with even- and odd-numbered license plates.

The measurement site as a typical residential area is surrounded by several main roads, and the

nocturnal traffic emission is relatively stable. Because NO was not measured at ground level after 14:00 on December 10$^{th}$, the direct HONO emissions during the third episode (E3, December 11$^{th}$ to 12$^{th}$) cannot be directly estimated. The contribution of direct emissions to ambient HONO during the first episode (E1, from December 7$^{th}$ to 10:00 on December 8$^{th}$) and the second episode (C2, from 10:00 on December 8$^{th}$ to December 11$^{th}$) were estimated to be 26% ± 5.9% and 32.8% ± 11.8%, respectively, which were comparable to the preceding results. Therefore, we used an average HONO$_{emis}$/HONO ratio to evaluate the relative contribution of direct HONO emissions during E3, which was used to correct observed HONO concentration in the subsequent analysis.

As the referee suggested, the budget analysis of nighttime HONO has been utilized to estimate the sources/sinks of HONO for the nocturnal profiles at different pollution periods. The nocturnal production of HONO on aerosol and ground surfaces have different relative contribution at different pollution periods, as the referee expected. We have significantly revised the discussion in the manuscript in "3.3 Direct emissions", "3.4.2 Relative important of aerosol and ground surfaces in nocturnal HONO production" and "3.4.3 Nocturnal HONO production and loss at ground level" to address the referee's concerns and to make our argument more clearly.

*When determining the relative contribution of aerosol surfaces to the HONO budget, the authors attempt to quantify HONO production using an assumed uptake coefficient (for NO$_2$-to-HONO conversion) and an aerosol surface area-to-volume ratio from a completely different study, which is supposed to be a typical of winter in Beijing. I don't see how one can justify using a single aerosol surface area-to-volume ratio for the entire campaign, especially since this will vary with time, meteorology, and day (see Table 1). Further, it is not clear why a single value was used since a suite of aerosol measurements were made during the current study; measurements of black carbon, non-refractory PM, and AMS measurements at ground and 260 m height are reported. This data should be used to refine the analysis and make it more specific for each ascent/descent period. I encourage the authors to provide a more in-depth analysis of the contribution of aerosol surface area to HONO processes. I believe there may indeed be evidence for an important role for aerosol here under hazy conditions. For example, clear gradients, denoted by decreasing HONO/NO$_2$ ratio with elevation, were only observed on a few occasions, whereas it was more often the case that the HONO/NO$_2$ ratio was quite constant over the vertical range. Thus, it is possible that aerosol*

*chemistry may influence this profile, but only a detailed quantitative analysis can tell you this.*

**Response:** As the referee suggested, the newly obtained aerosol surface areas ($S_a$) at ground level and at 260 m were used to estimate the nocturnal production of HONO by heterogeneous conversion of $NO_2$ on aerosol surface. A hygroscopic factor was applied to correct $S_a$ to ambient (wet) aerosol surface area ($S_{aw}$) for particle hygroscopicity (Liu et al., 2013; Wang et al., 2018). The time series of $S_{aw}$ at ground level and at 260 m was shown in Figure. S2 in the supplementary information. The surface area information for particles larger than 0.5 µm were not valid at ground level and 260 m during the measurement period. Hence, this is a lower limit estimate of the total surface area for the heterogeneous reaction.

An estimate of the nocturnal HONO production on aerosol surface was made using the RH corrected aerosol surface area ($S_{aw}$) and $NO_2$ observations from the residual layer. The correlation between CO and BC observed at ground level and the CO and BC at 260 m were analyzed. The CO and BC measured at ground level were independent of CO and BC observed at 260 m during the haze period (Figure. S10), since it can be expected that air masses in the residual layer were decoupled from the ground-level processes and largely free of $NO_2$ emissions. (Brown et al., 2012; VandenBoer et al., 2013). The vertical profiles of HONO and $NO_2$, and aerosol surface areas measured at 260 m allowed to estimate the nocturnal HONO production by heterogeneous conversion of $NO_2$ on aerosol surface. The HONO production from the heterogeneous $NO_2$ conversion ( $2NO_{2(g)}+H_2O_{(ads)} \xrightarrow{surface} HONO_{(g)}+HNO_{3(ads)}$) on aerosol surface would then have become the primary HONO source in the residual layer during E3. The yield of hydrolysis reaction assumes that HONO and $HNO_3$ are formed by equimolar disproportionation of two $NO_2$ molecules and immediately release HONO (Finlayson-Pitts et al., 2003; Finlayson-Pitts, 2009). The reactive uptake of $NO_2$ by the aerosol was assumed to occur on all measured aerosol surface area, regardless of chemical composition. HONO production was calculated using the equation of Ye et al. (2018) modified to account for the disproportionation. The uptake coefficient of $NO_2$ ranged between $1\times10^{-5}$ to $1 \times 10^{-6}$ in the dark, an average $S_{aw}$ of 2314 $\mu m^2$ $cm^{-3}$ between 22 and 01 h (the vertical measurement periods on Decemebr 11[th]) and an upper limit of observed $NO_2$ of 54 ppb from the residual layer were used to estimate the heterogeneous production of HONO on aerosols during E3. The HONO production of 30–300 ppt in an interval of 1.5 h could account for the observed HONO increases of 15–368 ppt in

the residual layer between profile measurements. The heterogeneous production of HONO solely on aerosol surface explained the HONO observations during E3. As the referee expected, the aerosol surface plays an important role in nocturnal HONO production during the haze episode. The column average concentration of HONO was independent of the mixing ratio of HONO observed from the ground level to 10 m in height ($R^2 = 0.27$) during E3 (Figure 10a), which suggested that aerosols presumably dominated the production of HONO aloft by heterogeneous uptake of $NO_2$.

The production of HONO on aerosols were also discussed to the other vertical measurements during C2, as the referee #1 suggested. The column average HONO concentration was related to the mixing ratio of HONO observed from the ground level to 10 m (Figure 10b, $R^2 = 0.93$), suggesting that the surface HONO source affected the HONO observed throughout the depth of boundary layer during C2. A high correlations ($R^2 = 0.83$) between measured CO and BC at ground level and the CO and BC at 260 m were observed (Figure. S10), which indicated that vehicle emissions affected air masses in the residual layer. The lack of NO vertical profile cannot directly correct the influence of direct HONO emissions. If it is assumed that the contribution of direct HONO emissions (35.9% ± 11.8%) was consistent at ground level and in the residual layer, the relative importance of aerosol and ground surface in nocturnal HONO production could be roughly estimated. It is necessary to elaborate that the contribution of direct emissions to HONO levels in the residual layer is a higher limit estimate. The production of HONO on aerosol surface was estimated during vertical measurements on December 9[th] and 10[th]. The HONO increases of 305–608 ppt between the vertical profile measurements, which have the contribution from direct HONO emissions subtracted, were higher than the production of HONO (26–259 ppt) on aerosols in an interval of 5.5 h on December 9[th]. The production of HONO solely on aerosol surface cannot explain the observed HONO increases in the residual layer, suggesting that the observed HONO throughout the depth of boundary layer was primarily derived from the heterogeneous $NO_2$ conversion on the ground surface followed by vertical transport throughout the column. The HONO production from the aerosol surface in an interval of 5.35 h was 33–332 ppt in the residual layer on December 10[th], which was comparable to the corrected HONO increases of 114–369 ppt between profile measurements. This could have been due to a shallow inversion layer formed near the surface (Figure 4), which inhibited the vertical transport of nighttime HONO at ground level. Besides the contribution of direct HONO emissions could be even more overestimated, which might also affect the estimated result. All of these may result in the

aerosols dominated the heterogeneous production of HONO from $NO_2$ in the residual layer on December 10th.

As the referee expected, the aerosol chemistry plays an important role under hazy conditions. The production of HONO solely on aerosol surface explained the HONO observations during the haze episode. However, the ground surface was still the dominant nocturnal surface on which HONO was formed from the heterogeneous conversion of $NO_2$ during the clean episode. We have significantly revised the discussion in section 3.4.2 as suggested, adding more detailed quantitative analysis to draw broad conclusions.

*Finally, the authors state that ground level HONO production has been corrected for the influence of direct emissions from automobiles. Has this correction been carried out for the production on aerosol surfaces as well? I had a difficult time understanding the rationale of going through all the calculations in section 3.4.3 on the topic of ground level HONO production. It was not clear how equation 4 and 5 were derived. The purpose of these equations is to estimate a HONO conversion frequency, which appears to be the pseudo-first order rate constant associated with the conversion of $NO_2$-to-HONO. These values should then be used to compare to the other HONO sources and sinks to evaluate the relative importance of all the various HONO sources/sinks. However, this comparison is not clearly carried out at the end of the paper as I would expect. Instead, starting with line 410 the text veers off subject by deriving a HONO yield from deposited $NO_2$. While this may be a useful calculation to carry out, I don't see how it helps the authors reach their research objectives and main conclusions.*

*In closing, I feel the interpretation of the vertical profiles relies on highly generalized 'back of the envelope' calculations. Such calculations can be very useful in certain cases; however, the quality and quantity of the data provided by this study should allow for a more elaborate analyses. I recommend the authors revisit their data to devise a mass balance for HONO processes where they quantify the sources and sinks for each vertical profile. I expect this approach will provide valuable insights that are not clear from the current presentation of the data.*

**Response:** Thanks for the comment and suggestion. The influence of direct HONO emissions was taken into account in evaluating the heterogeneous production of HONO on aerosol surface, which has been discussed in section 3.4.2 of the revised manuscript. Equation 4 and 5 were derived by Su et

al. (2008) to correct for the effects of source and diffusion, which was derived from the equation ($[Y]_t^{Xscaled} = \overline{[X]} \frac{[Y]_t}{[X]_t}$), in which the mixing ratio of $Y$ at time $t$ was scaled with species $X$ at the same time (Lammel, 1999). As the referee suggested, the slope determined by linear regression will statistically better describe the efficient first order conversion rate constant of $NO_2$ to HONO. Thus the conversion frequency was re-estimated following the method by Alicke et al. (2002). The conversion frequencies ($k_{HONO}$) on December 9[th], 10[th], and 11[th] were 0.0082, 0.0060 and 0.0114 h[-1], respectively, corresponding to a HONO production rate by $NO_2$ ($P_{NO_2}$) of $0.25 \pm 0.03$, $0.28 \pm 0.02$ and $0.60 \pm 0.02$ ppb h[-1] (i.e. $C_{HONO} \times \overline{[NO_2]}$). It is necessary to elaborate that the derived $P_{NO_2}$ is net HONO production, which means sources and sinks of HONO (aerosol and ground surface source, deposition, etc.) have already been taken into account in $P_{NO_2}$.

As the referee suggested, the surface production rate of HONO was used to compare to the production rate on aerosols based on the assumption that the production of HONO on aerosols was insignificant compared to the ground surface during the clean episode, which has been suggested in other studies of HONO vertical gradient (VandenBoer et al., 2013; Wong et al., 2011; Zhang et al., 2009). The $P_{NO_2}$ could be approximately considered as the net contribution of surface HONO production to the column during C2. The production rate of HONO on the ground surface ($0.25 \pm$ 0.03 and $0.28 \pm 0.02$ ppb h[-1]) was an order of magnitude higher than the maximum production rate on aerosol surface (0.047 and 0.062 ppb h[-1]), which suggested that the HONO observed throughout the column was primarily derived from the heterogeneous conversion of $NO_2$ on the ground surface followed by vertical transport throughout the column during the clean episode. This result was consistent with the result in section 3.4.2. In contrast, the production of HONO on aerosols as an important nighttime HONO source cannot be ignored during the haze episode, as discussed in section 3.4.2. To compare the production of HONO on aerosol and ground surfaces, the production rate of HONO on the ground surface was estimated by equation (Eq. 8) with a deposition velocity ($V_{dep,NO_2}$) of 0.07 cm s[-1] (VandenBoer et al., 2013) and a boundary layer height of 140 m. The derived production rate of HONO ($P_{HONO,ground}$) on the ground surface was $0.47 \pm 0.02$ ppb h[-1] on December 11[th] (E3), which was comparable to the production rate on aerosol surface of 0.2 ppb h[-1]. This result also suggests that production of HONO on aerosols is an important nocturnal source of HONO during the haze episode.

A budget analysis of nighttime HONO was utilized to evaluate the contributions of various HONO sources/sinks, as the referee suggested. The derived a net HONO yield from the ground surface conversion of $NO_2$ have been removed in the revised manuscript. The nocturnal HONO budget from 18:00 to 06:00 LT on December 9th (C2) and 10th (E3) were shown in Figure 11 in the revised manuscript. The average production rate of HONO on aerosols ($0.04 \pm 0.01$ ppb h$^{-1}$) was insignificant compared to the surface production rate of $0.28 \pm 0.03$ ppb h$^{-1}$ during the clean episode. However, the average $P_{aerosol}$ of $0.19 \pm 0.01$ ppb h$^{-1}$ was comparable to the surface production rate of HONO ($P_{ground}$, $0.47 \pm 0.03$ ppb h$^{-1}$) during the haze episode. Direct HONO emissions just contributed a small portion of HONO at a rate of $0.06 \pm 0.07$ and $0.10 \pm 0.10$ ppb h$^{-1}$ during C2 and E3, respectively. The dry deposition of HONO contributed $0.74 \pm 0.31$ and $1.55 \pm 0.32$ ppb h$^{-1}$ to the nocturnal loss of HONO during C2 and E3, respectively, implying that significant quantities of HONO were deposited to the ground surface at night. This had been suggested in another study on the vertical gradient of HONO (VandenBoer et al., 2013).

**Specific comments by line number:**

*Lines 45-133: I feel that the introduction lacks an explicit statement of the study's research objective(s). The authors provide a [somewhat long] background section reviewing aspects of HONO chemistry that ends with a summary of what they did in their study. The introduction would be far stronger if the authors include a focused discussion of what research objective they hope to achieve. This could be accomplished by stating a hypothesis, following by a plan for how they hoped to test the hypothesis. At the moment, it is only in the second to last sentence of the introduction that one finally learns that a comparison of HONO heterogeneous chemistry on ground vs. aerosol surfaces is the major aim of the work.*

**Response:** We have significantly revised the discussion in the introduction section. The background introduction of HONO chemistry and vertical observations have been revised as suggested, and more explicit statements of research objective have been added to the introduction section.

*64-66: The authors state that hydrolysis of $NO_2$ on humid surfaces via R2 is considered the most likely explanation for observed HONO concentrations. The authors should be aware that this is not a viable mechanism under atmospherically relevant concentrations of $NO_2$. Reaction R2 is too slow and only*

*becomes important when the NO$_2$ concentration is high enough to promote N$_2$O$_4$ formation. Initial mechanistic studies of this mechanism had to use high NO$_2$ concentrations due to high instrument detection limits. More recent laboratory studies of this process using state-of-the-science sensitive field-grade instrumentation do not report this chemistry occurring under lower concentrations (50 ppb or less). One can refer to numerous lab studies over the past 10-15 years aimed at studying NO$_2$ conversion on surfaces containing humic acid reactive sites to learn more about this. It is notable that these studies often include control experiments aimed at quantifying background levels of HONO derived from NO$_2$ hydrolysis on their experimental system (i.e., in the absence of redox active substrates). These studies consistently show that NO$_2$-to-HONO conversion on organic matter and other redox active surfaces produce orders of magnitude more HONO than does R2.*

**Response:** We would like to thank the referee for the constructive suggestion. The introduction has been significantly revised as suggested. This statements have been revised as below,

**Line 86–92**

"The heterogeneous reduction of NO$_2$ with organic substrates is proposed to be another effective pathway to generate HONO (Brigante et al., 2008; Stemmler et al., 2006; George et al., 2005). However, extrapolation of lab results to real surfaces remains challenging. The nocturnal production of HONO has been considered to be dominated by the NO$_2$ heterogeneous reaction (R2). Although the heterogeneous reaction (R2) of HONO formation is first-order in NO$_2$, the mechanism for the conversion of NO$_2$ on surfaces remains unclear (Finlayson-Pitts et al., 2003; Finlayson-Pitts, 2009)."

*81: The colon after the references should be deleted.*

**Response:** Revision has made as the referee suggested.

*349: The authors attribute a decrease in HONO/NO$_2$ ratio at RH above 70% to enhanced HONO deposition. While RH does impact HONO uptake (see Donaldson et al. Environ. Sci. Technol. 2014, 48, 375), it would affect NO$_2$-to-HONO conversion as well (by enhancing to a point); so there are competing effects here. The abovementioned Donaldson et al. paper suggests that the uptake coefficient for HONO actually decreases with increasing RH.*

**Response:** Thanks for the comment and suggestion. We have revised the description in this paragraph, as below.

**Line 548–558:**

"A previous observation at ground level also reported that the HONO/NO$_2$ ratio increased with an increase in RH when the RH was less than 70%. A further increase in RH would lead to a decrease in the HONO/NO$_2$ ratio, which was considered to be caused by the number of water monolayers that formed on the surface leading to an efficient uptake of HONO (Li et al., 2012; Yu et al., 2009; Liu et al., 2019). However, a decreased uptake coefficient of HONO with increasing RH from 0% to 80% was observed in a laboratory study (Donaldson et al., 2014). The NO$_2$ to HONO conversion efficiency depended negatively on RH at an RH above 70%, which was presumably caused by the relative humidity affecting both HONO uptake onto the surface and the NO$_2$-to-HONO conversion."

*372: The number listed having unit of h$^{-1}$ is not a formation rate as stated. Those are units of a first-order rate constant. Rather, units of ppb h$^{-1}$ are appropriate for a formation rate.*

**Response:** The unit of the NO$_2$-normalized HONO production over time, $\Delta \frac{[HONO]}{[NO_2]}/\Delta t$, is h$^{-1}$ according to the equation (Eq. 5). The unit of its equivalent HONO production rate at constant NO$_2$ concentration is ppb h$^{-1}$. We have revised the statements in this paragraph to make it clearer.

*405: It would be beneficial to provide a visual comparison of all the production rates (calculated in units of mixing ratio per hour). See for example Fig. 11 from Vandenboer et al. JGR, 2013, 118, 10,155.*

**Response:** As the referee suggested, the comparison of the production and loss rate of HONO have been added in section 3.4.3. Figure 11 shows the production and loss rate of nighttime HONO on December 9$^{th}$ and 10$^{th}$.

*413: I recommend that the authors choose another symbol to represent the compensation point, which they call the HONO yield from deposited NO$_2$; the greek letter phi is reserved for the photolysis quantum yield. The symbol "H" used for mixing depth is also confusing. Some readers may mistake this for the Henry's law constant without reading further into the text. Consider using "d" (for depth) or lower case "h" or "z", all of which have been used in the past to represent height.*

**Response:** Following the referee's suggestion in the specific comments above, the derived a net

HONO yield from ground surface conversion of $NO_2$ have been removed in the revised manuscript. The symbol "*H*" used for mixing height has been modified to symbol "h" as the referee suggested.

*421: Why choose a deposition velocity from the literature, which is derived from completely different studies. Isn't the point here to model the vertical profile to derive a deposition velocity and the uptake coefficients for $NO_2$, HONO, etc. that are specific for the current sampling site?*

**Response:** As the referee suggested, the temperature-dependent deposition velocity of HONO ($V(HONO)_T = \exp(23920/T - 91.5)$), which was adjusted to value of 2 cm s$^{-1}$ at 0 °C decreasing exponentially to non-significant values at 40 °C (Laufs et al., 2017), was used to derive the dry deposition velocity of HONO. The $V_{dep,HONO}$ calculated from nocturnal observations (00:00–06:00 LT) ranged between 0.9 and 3 cm s$^{-1}$, with an average value of 1.8 cm s$^{-1}$, which was within the range of previously reported values (0.077–3 cm s$^{-1}$) (Harrison and Kitto, 1994; Harrison et al., 1996; Spindler et al., 1998; Stutz et al., 2002; Coe and Gallagher, 1992; Laufs et al., 2017). However, the limited vertical profile data in this study restricted the estimaton of the uptake coefficients for $NO_2$ and HONO. Thus we used the uptake coefficients and the deposition velocity from the literature to estimate the heterogeneous production of HONO on aerosol and ground surfaces, which have also been used in other studies of HONO vertical structure (VandenBoer et al., 2013; Wong et al., 2011; Ye et al., 2018).

**More general remarks:**

*More general remarks: More work should be devoted to providing error analysis. For example, error bars on data and errors in reported values derived from their analyses. This will be especially important when comparing quantitatively the relative contribution of sources/sinks to ambient HONO. I felt there were too many figures in the text. For example, what is the purpose of showing Figure 5, where the information is more useful plotted in the format shown in Fig. 6, etc. The manuscript could benefit from editing by a native English speaker. Grammatical errors start on the first line of the abstract and persist throughout the manuscript. Lastly, I note that the authors are comparing their IBBCEAS results to data collected on an instrument from Cambridge University. Does this contribution need to be acknowledged in some way (e.g., in the acknowledgement section or author list)?*

**Response:** Thanks for the comment and suggestion. The error analysis have been taken into account in data analysis, and error bars have been presented in the revised manuscript. Figures 5 have been removed from the manuscript as suggested, and the panel (a) of Figure 6 has been moved to the supplementary information. As the referee suggested, the revised manuscript was edited by a native English speaker. We also acknowledged Bin Ouyang from Cambridge University for providing HONO comparison data in the acknowledgement section.

**References:**

[revised manuscript text omitted]

---

## Author Response (AR2)

**Reply to comments from Referees**

We would like to thank the editor and two anonymous referees whose comments helped us improve the manuscript. Below we give a point-to-point response to address the referees' comments. The original comments are in blue italics and our responses are in black. The corrections are marked as red color in the attached revised manuscript.

**1. Response to Anonymous Referee #1**

*The Authors have done an extremely commendable revision to this manuscript. Despite the highly limited nature of their observational dataset, they have derived insight into some of the competing HONO mechanisms in the Beijing atmosphere under haze and clear pollution conditions. Pending some minor revisions and clarifications, the paper is acceptable for publication in Atmospheric Chemistry and Physics.*

**Minor Comments:**

*1. The Authors state in a number of places throughout the manuscript that aerosol production of HONO under haze conditions is dominant. This is incorrect. It is significant, but in the comparison against the ground source throughout the nocturnal boundary layer, aerosol production still only accounts for one third of the total HONO produced. This estimate is made using the upper limit for $NO_2$ reactive uptake from the literature, so it is likely less significant that reported and more in line with a wealth of prior studies. The Authors need to do a better job of performing their calculations using upper and lower limits instead of just an upper limit estimate. A conservative calculation or one which includes the variability in the observations is more reliable. Specific comments below provide some areas where this improvement is required to produce a more balanced interpretation of the dataset. Given the very limited nature of the observations, such analysis is likely to provide more robust insight for comparison when more targeted measurements are made in the future, raising the impact of this work.*

**Response:** Thank you very much for your valuable comments and suggestions. We have revised the manuscript according to the referee's specific comments below. Considering the upper and lower limits of uptake coefficient of $NO_2$ as suggested, the uptake coefficient of $5 \times 10^{-6}$ have been taken into account in the calculation and interpretation of the dataset. The aerosol production of HONO is an important nocturnal HONO source during haze episodes. The aerosol surface dominates HONO production aloft by heterogeneous uptake of $NO_2$ during haze episodes, and the surface production of HONO and direct emissions into the overlying air are minor contributors. The statements have been revised as the referee suggested.

*2. In Section 3.2 the Authors discuss the decreasing slopes determined using orthogonal least squares analysis. First, why the orthogonal approach was used is not clear, but presumably this was used as an extension from the revised intercomparison of the HONO measurements. The error in the measurement height must be quite a lot smaller than in the HONO measurement, which means that a least-squares regression is more likely appropriate. In either case, no examples of the regression line through the data are provided, nor are the parameters used to constrain the regression line. This limits the ability of bias in these trends to be identified. Were regressions performed through HONO data only above the surface layer? There is an exponential decrease in HONO with altitude within the surface layer that would be incorrect to apply a linear trend to, where the lowest HONO observation having the highest mixing ratio would significantly affect the slope of the line. The Authors need to clarify this approach in the manuscript so it can be compared to in future studies.*

**Response:** Thanks for the comment and suggestion. As the referee suggested, we used the linear squares regression of HONO and $NO_2$ to altitude to evaluate the nocturnal gradient of HONO and $NO_2$. Vertical profile data were used above the surface layer and 10 m vertical average from the surface to 240 m above ground level (AGL) to constrain the regression line, which has been clarified in the revised manuscript. An example of the regression line of HONO and $NO_2$ were added in the supplementary in formation as suggested. The alterations in the values of the slopes were revised throughout the manuscript.

*3. The Authors report 1-simga limit of detection (LOD). This is a gross error and misrepresentation of instrumental capability. Detection limits are standardly reported at 3-sigma which results in measurements with approximately 50 % error, decreasing quickly to 10 % error (or less) at the 10-sigma threshold (limit of quantitation). Report the 3-sigma LOD and indicate where this threshold*

*lies for each instrument on the Figures associated with the intercomparison and also on any plots of HONO mixing ratios where measurements are below the 3-sigma LOD. The Authors need to revise their error assigned to data between the 3-sigma and 10-sigma levels to be at least +/- 20 % of the measured value, while data below the 3-sigma LOD should be reported with +/- 100 % uncertainty. Finally, while the errors of the HONO measurements from the IBBCEAS are quantified nicely, the accuracy from the intercomparison is not actually reported. Once the LOD error re-analysis has been completed, then the slopes from the resulting intercomparisons should give a good estimate of between-instrument accuracy.*

**Response:** Thanks for the comment and suggestion. As the referee suggested, the 3σ limit of detection (LOD) of IBBCEAS instrument was reported in the manuscript. The errors assigned to data between the 3σ and 10σ were modified, and Figure 2 was also revised as suggested. As the referee pointed out, we re-analyzed the LOD error and reported the relative accuracy of comparison between the different instruments in the manuscript.

**Detailed Comments:**

*1. Lines 34-37: The results of this analysis need to be updated to reflect the relative importance under haze and clean conditions according to comments below.*

**Response:** We have updated and modified this analysis as the referee suggested.

*2. Lines 49-51: The references used in this sentence appear to be placed backwards. Swap them.*

**Response:** Revision has made as the referee suggested.

*3. Line 87: Should be 'Americas'*

**Response:** Revision has made as the referee suggested.

*4. Line 118: Should be '…aerosol surfaces play a…'*

**Response:** Revision has made as the referee suggested.

*5. Lines 161-162: Please provide an estimate of the total path length.*

**Response:** An estimate of the total optical path length has been added in the manuscript as suggested.

*6. Lines 164-174: Revise according to Minor Comment 3. Be very clear in how the 3-sigma LOD was calculated so it can be used for comparison by others who also measure HONO by IBBCEAS. It is too bad that a standard amount of gas phase HONO was not delivered to these systems to ensure that inlet or cavity-sorption losses of HONO were not changing over time and impacting the accuracy of the measurements.*

**Response:** Thanks for the comment and suggestion. We have calculated the $3\sigma$ LOD of the IBBCEAS instrument as suggested, which has been modified in the manuscript. As the referee pointed out, a HONO standard generator was used to determine the sample loss of HONO. In the experiment, the RH was about 65% and the temperature was about 23 °C, the sample loss of HONO was found to be ~2.0% as shown in Figure 1. The variation of HONO sample loss over time was negligible. The similar approach was used to determine the sample loss of $NO_2$ and found that it was negligible. A more detailed description of the IBBCEAS instrument can be found in our published manuscript (Duan et al., 2018).

[Figure]

**Figure 1.** The black circles correspond to the HONO observed from the HONO standard generator. The red circle correspond to the HONO measured with the extra 1 μm PTFE filter, 3 m PFA inlet tube and the simulative PFA optical cavity tube added in front of the IBBCEAS instrument.

*7. Lines 178-183: This section needs clarification. Were these measurements made at all heights? Only on the ground? Or were these instruments located in the moving basket?*

**Response:** Thanks for your comments. CO and $O_3$ were measured simultaneously at ground level and at 260 m on the tower, while NO was measured only at ground level. We have made the clarification in the manuscript as the referee suggested.

*8. Line 184: In comparison to what? Clarify.*

**Response:** We have removed this sentence and added the accuracy of the instrument in the manuscript to make our statement more clearly.

*9. Line 202: Please add why use of these a and b terms are reasonable to use. The paper cited is for measurements made much further south, in the Pearl River Delta. A sound justification for the use of aerosol parameters from a location so far away, with far more agricultural impact on particulate composition (also different typical T and RH).*

**Response:** As the referee suggested, we have made the clarification in the manuscript. The parameters *a* and *b* were derived using the measurement dataset from Guangzhou region, which, like Beijing, is one of the mega-cities in China. The curve-fitting parameters *a* (2.06) and *b* (3.6) for urban areas were used in the manuscript.

*10. Lines 213-219: Use all of these intercomparisons to determine some measure of relative accuracy between these four instruments (i.e. the relative standard deviation of the slopes).*

**Response:** We have modified this section and added the relative measurement difference between the four instruments.

*11. Line 235: 0.05 ppb is below the 1-sigma LOD. These low mixing ratios cannot be reported with any reliability. The Authors need to reconsider their interpretation anywhere that measurements are below 3-sigma LOD.*

**Response:** Thanks for the comment and suggestion. We have revised the manuscript as the referee suggested. The measurements below 3σ LOD have been removed in the manuscript.

*12. Line 241: Delete 'other'*

**Response:** Revision has made as the referee suggested.

*13. Line 244: Which measurement or period these numbers correspond to is not clear. Please revise for clarity.*

**Response:** The sentence have been revised to make our statement more clearly.

*14. Line 264: +/- 0.4 ppb is very close to the 3-sigma detection limit, which suggests that these reported values may only be an evaluation of the difference in noise between the two instruments.*

**Response:** We have revised the discussion as the referee suggested.

*15. Lines 331-347: Apply corrections for Minor Comment 2 here and make changes throughout manuscript to match any alterations in the values of the slopes and their interpretation.*

**Response:** Revision has made as the referee suggested

*16. Lines 470-472: This should be calculated using the lower limit of the $NO_2$ mixing ratio (~40 ppbv) to provide a conservative estimate of the production rate. This reads as if the upper limit was used because it conveniently matched the observations, yet all of the remaining $NO_2$ measurements are below this value, which indicates that the aerosol surface conversion is a major contributor in isolated air parcels, but not the only contributor. This is more internally-consistent with the findings presented later in the manuscript where ground and aerosol surface contributions are compared throughout two nocturnal boundary layers after direct emissions are accounted for.*

**Response:** Thanks for the comment and suggestion. We calculated the HONO production using the average $NO_2$ mixing ratio (52.88 ppb) observed in the residual layer during E3. The absolute amount produced in an interval of 1.5 h could match the observations between vertical profile measurements (time interval: 1.5 h). As the referee pointed out, the aerosol surface dominates HONO production aloft by heterogeneous conversion of $NO_2$ during haze episodes, and surface production of HONO and direct emissions into the overlying air are minor contributors.

*17. Lines 472-474: This is a comparison between the calculation and the observations, yet it is not clear which numbers belong to the calculation and the observation here. Also, the Authors should normalize to a constant time interval (e.g. pptv per hour) for both the calculation and the observation numbers, since one interval given here is 1.5 hours and the other is 'between two vertical profile*

*measurements'. There are other instances of comparisons between calculations and observations from here onward that make following the results difficult, if not impossible.*

**Response:** Thanks for the comment and suggestion. As the referee suggested, we have revised this section to make our argument more clearly. The constant interval time for both the calculation and the observation was used for comparison between the calculation and the observation. The time interval of 1.5 h used in the calculation is same as the time interval between two vertical profile measurements (1.5 h). The statement throughout the section 3.4.2 have been revised to make our argument more clearly.

*18. Lines 481-482: The end of this sentence should be modified to indicate that ground production of HONO or direct emission into the surface layer are minor contributors, according to the results of the addressing the prior comments in this section.*

**Response:** Revision has made as the referee suggested.

*19. Line 487: The Authors should again emphasize the severe limitation of the number of transects in their dataset here. This haze event has not been characterized for how typical or atypical it may be in terms of chemical composition and so these results may not hold under a more thorough investigation of HONO formation in Beijing during other haze episodes, yet provides motivation to find out!*

**Response:** As the referee suggested, we have emphasized the limitation of the number of vertical profile data in our study.

*20. Lines 501-507: Upper limit used again here, should use lower limit. Calculated formation rates are in ppbv/hr while HONO increases are pptv between transects. Please use the same units for both values to make the comparison simple for your readers. Also clearly state what fraction of the total HONO production these calculated rates account for.*

**Response:** Thanks for the comment and suggestion. The average concentration of $NO_2$ observed in the residual layer were used in the calculation. As the referee suggested, we have revised the discussion of the comparison between the calculation and the observation to make our argument more clearly. The fraction of the calculated HONO production to the total HONO have been added in the manuscript as suggested.

*21. Lines 510-516: The clarity of writing here makes this challenging to follow. Given the limitations given in the discussion here, it would seem that this vertical transect was not a very good case study for analysis. It may be better to remove this from the discussion.*

**Response:** Thanks for the comment and suggestion. We have revised this section as the referee suggested.

*22. Lines 521-522: The Authors state 'presumably dominated', yet they performed a quantitative assessment. Give the calculated fraction of the observed total HONO production. Clearly state that it only applies for a limited number of transects over a short period of a given night.*

**Response:** The sentence has been revised as the referee suggested.

*23. Lines 546-547: The Zhang et al (2018) observation in Beijing during haze has a value much higher than given here. Is this a typo? If not, then this much higher observation should be moved to the next sentence. A period should precede 'However' as well, instead of a comma.*

**Response:** As the referee pointed out, this is not a typo. We have moved it to the next sentence as the referee suggested.

*24. Line 559: The Authors are creating confusion here. The aerosol surface conversion of HONO in the residual layer dominated its production in that layer, but this does not tell us anything about how important aerosol conversion is throughout the nighttime troposphere until it is compared in the following discussion of surface production. Revise this here to 'is an active HONO production mechanism during haze episodes' to improve accuracy and clarity.*

**Response:** Revision has made as the referee suggested.

*25. Line 568: Revise to use the lower limits calculated and/or give the mean with +/- corresponding to reach the upper and lower limits. This will give a much better picture of the chemistry in this very nice comparison section of the discussion.*

**Response:** Thanks for the comment and suggestion. As the referee suggested, we gave the upper limit and the lower limit, and the mean with +/- corresponding to reach the upper and lower limits.

*26. Line 598: Again, only an upper limit is used here, yet a reactive uptake coefficient of 10^-6 is common to find in the literature. The lower limit or the entire range should be explored. The upper limit use here, again, suggests the Authors are more interested in the upper limit of aerosol conversion importance rather than providing a balanced perspective. Given their extremely limited dataset, caution in the interpretation of this data will make the results of this work more valuable for future comparisons. Please revise here and for the calculations moving forward through the remaining discussion.*

**Response:** Thanks for the comment and suggestion. As the referee suggested, we considered the upper and lower limits of the uptake coefficient ($1\times10^{-5} - 1\times10^{-6}$), the uptake coefficient of $5\times10^{-6}$ was used in the calculation. We have modified the calculations here and in the discussion below, as the referee suggested.

*27. Lines 609-610: And when the reactive uptake of $NO_2$ is 10^-6, then how does it compare? The Authors should provide the fraction of aerosol HONO production to the total calculated here to indicate its importance. It is only one third of the total, at the upper limit, which means it is important, but not dominant.*

**Response:** Thanks for the comment and suggestion. Considering the upper and lower limits of the uptake coefficient of $NO_2$, the uptake coefficient of $5\times10^{-6}$ was used in the calculation. As the referee suggested, we have provided the fraction of aerosol production of HONO to the total production of HONO, contributing about 20% of the production of HONO, suggesting that aerosol production of HONO is an important nocturnal HONO source.

*28. Line 619: Revise this section according to manuscript changes throughout.*

**Response:** As the referee suggested, we have revised the conclusion according to the changes in the manuscript.

**General comments:**

*I feel that the authors have satisfactorily addressed most of my concerns. The only comment I have is regarding equation (4) and (5) on p. 16. The Authors state "the yield of the hydrolysis reaction assumes that HONO and $HNO_3$ are formed by equimolar disproportionation of two $NO_2$ molecules and immediately release HONO." (line 454-456) They further state that equations (4) and (5) have been modified to account for the disproportionation. However, the $NO_2$ uptake coefficients used are independent of any product yield for reaction (R2). I recommend deleting the above quoted sentence on line 454-456 since it does not have any bearing on equation (4) and (5). In fact, as mentioned in my first review of this paper. It is well established in the literature that hydrolysis of $N_2O_4$ only occurs at ppm levels of $NO_2$, when the $2NO_2 = N_2O_4$ equilibrium favors $N_2O_4$ formation; those concentrations are not present in the air of Beijing.*

**Response:** Thanks for the comment and suggestion. The quoted sentence have been removed as the referee suggested.

[revised manuscript text omitted]

---

## Author Response (AR3)

**Reply to comments from Editor**

We thank the editor for her editorial work and are glad that this paper was accepted for publication. Below we give a point-to-point response to address the editor comments. The original comments are in blue italics and our responses are in black. The corrections are marked as red color in the attached revised manuscript.

**Response to Editor**

*I commend the authors on their substantial improvements to the manuscript. I have some minor suggestions for technical corrections (below) but I do not need to review the paper again before publication in ACP.*

**Comments:**

*1. The period on line 114 should be a comma because the second sentence is actually just the last part of the previous sentence.*

**Response:** Revision has made as the editor suggested.

*2. Lines 166 – 170 – Why are two different ranges being used to describe the detection limits? Is it because the first set are for 30 sec averages and the second set are for 15 sec averages? I suggest that the authors move the information about the 3-sigma detection limit for 30 sec averages (line 166-167) to after the sentence ending on line 172 about the ground-based system to avoid confusion.*

**Response:** Thanks for the comment and suggestion. As the editor pointed out, we took the two IBBCEAS instruments to measure the HONO and $NO_2$, which were respectively mounted in the container and on the ground. The time resolution of the two IBBCEAS instruments was set to 15s and 30s, respectively, with different detection limits. We have moved the sentence to after the sentence ending on line 172 to avoid confusion, as the editor suggested.

*3. Line 193 Aerodyne should be capitalized*

**Response:** Revision has made as the editor suggested.

*4. Figure 3 – please clarify what height the NO, NO₂ and HONO measurements are from in this figure*

**Response:** As the editor suggested, we have clarified the measured height of NO, NO$_2$ and HONO in Figure 3.

*5. Figure 7 – I find a pie chart a very unusual choice to display this data. The authors can keep it if they prefer, but I think the data would make a lot more sense displayed as a histogram with bars indicating the frequency of the ratio of HONO$_{emit}$/HONO in increments of 0.1 (i.e. 10%).*

**Response:** Thanks for the comment and suggestion. We decided to keep the pie chart because it shows the frequency distribution of different HONO$_{emis}$/HONO ratios. The figure shows the low HONO$_{emis}$/HONO ratio dominating the frequency distribution, which indicated that direct HONO emissions contributed only part of the nocturnal HONO concentration.

*6. Equation 5 – The authors don't mention it, but this version of the equation relies on the assumption that the uptake of NO₂ to form HONO doesn't cause a significant change to the NO₂ concentration over the time period of interest. This assumption appears very valid here, but I think it's worth mentioning this explicitly so that if future researchers want to apply this method, they understand the limitation or caveat.*

**Response:** Thanks for the comment and suggestion. As the editor suggested, we have added the assumption of equation 5 to the revised manuscript.

*7. Author contributions – I find the last sentence in the author statement to be confusing. You have clearly outlined the separate contributions of FM and MQ, so why say that they are 'the same'?*

**Response:** We have revised this sentence to make our statement more clearly. We changed it to "
[revised manuscript text omitted]